# Im-Promptu: In-Context Composition
# from Image Prompts

**Bhishma Dedhia[1], Michael Chang[2], Jake C. Snell[3], Thomas L. Griffiths[3,4], Niraj K. Jha[1]**

[1]Department of Electrical and Computer Engineering, Princeton University
[2]Department of Computer Science, University of California Berkeley
[3]Department of Computer Science, Princeton University
[4]Department of Psychology, Princeton University
{bdedhia,js2523,tomg,jha}@princeton.edu, mbchang@berkeley.edu

## Abstract

Large language models are few-shot learners that can solve diverse tasks from a handful of demonstrations. This implicit understanding of tasks suggests that the attention mechanisms over word tokens may play a role in analogical reasoning. In this work, we investigate whether analogical reasoning can enable in-context composition over composable elements of visual stimuli. First, we introduce a suite of three benchmarks to test the generalization properties of a visual in-context learner. We formalize the notion of an analogy-based in-context learner and use it to design a meta-learning framework called *Im-Promptu*. Whereas the requisite token granularity for language is well established, the appropriate compositional granularity for enabling in-context generalization in visual stimuli is usually unspecified. To this end, we use *Im-Promptu* to train multiple agents with different levels of compositionality, including vector representations, patch representations, and object slots. Our experiments reveal tradeoffs between extrapolation abilities and the degree of compositionality, with non-compositional representations extending learned composition rules to unseen domains but performing poorly on combinatorial tasks. Patch-based representations require patches to contain entire objects for robust extrapolation. At the same time, object-centric tokenizers coupled with a cross-attention module generate consistent and high-fidelity solutions, with these inductive biases being particularly crucial for compositional generalization. Lastly, we demonstrate a use case of *Im-Promptu* as an intuitive programming interface for image generation.

## 1 Introduction

> No thought can be formed that isn't informed by the past; or, more precisely, we think only thanks to analogies that link our present to our past.

D. Hofstadter and E. Sander, Surfaces and Essences [1]

Humans represent complex concepts by combining and organizing simpler concepts [2, 3]. This endows us with an uncanny skill for constructing an unlimited number of concepts by composing a relatively small set of previously learned basic building blocks, an ability more formally called *compositional generalization*. For example, we can generate sentences by combining a dictionary of words in different ways and solve a wide range of mathematical problems using a small set of basic arithmetic operations and variables. In the case of the visual world, objects form naturally

37th Conference on Neural Information Processing Systems (NeurIPS 2023).

composable entities and object-centric primitives can be flexibly combined using abstract syntactic rules to yield novel and counterfactual scenes.

The growing body of work in object-centric learning methods [4, 5, 6] enable the extraction of latent object representations from perceptual scenes. However, slot composition methods have been limited to *ad-hoc* composition rules based on hard-coding slots into concept libraries [7, 8]. A central challenge in learning a generative model that explicitly models the compositional grammar of the visual world is that the seemingly unlimited number of composition rules that undergird real-life visual entities prevents the *explicit* specification of each rule. For example, a warm spring day consists of blooming flowers and freshly verdant expanses, but come autumn, the same scene stipulates the presence of golden meadows carpeted with fallen leaves. State-of-the-art text-to-image models [9, 10, 11, 12] circumvent this bottleneck by grounding images in natural language. This aligns the representations of entities in an image to the compositional structure of language. While the visual quality of these models is stunning, they often require engineering complex language prompts to elicit the desired output. They also suffer from limited understanding of object relations [13]. Moreover, studies [14] have shown that infants as young as six months old can extract compositional structure from sequences of visual events, even before they acquire language. Such compositional abstraction independent of language faculty exhibited by humans remains elusive for contemporary models.

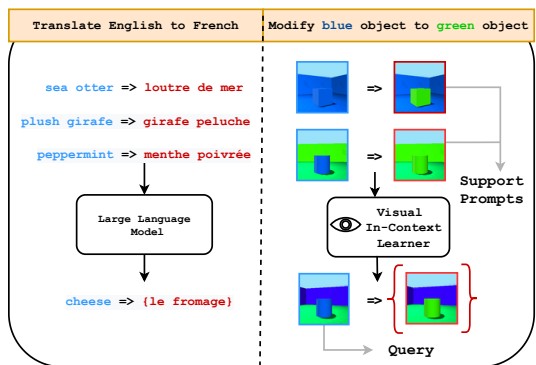

Figure 1: LLMs perform in-context understanding of a task from a few examples in an analogical manner (left). We use analogy-making to implicitly understand the composition rules over visual stimuli made of object-like entities (right).

Large language models (LLMs) [15, 16, 17, 18, 19, 20], on the other hand, demonstrate *implicit* composition skills. Pre-trained on large text corpora, these models have impressive emergent few-shot learning capabilities. They can solve novel tasks on the fly without gradient updates by following examples shown in the instruction prompts, a process known as *in-context learning*. Under the hood, in-context learning uses the inherently compositional nature of the input text to implicitly infer the task structure and how to apply the task structure to a query [21]. In the most simple form, the prompt-based few-shot learning paradigm can be formalized as *analogy solving* of the type $A : B :: C : D$, where $A$ represents an example task, $B$ represents the example solution, and $C$ denotes the query. The generated solution is the analogy completion $D$. Analogies over compositional elements, therefore, provide a direct isomorphism between in-context learning for language and learning implicit composition rules for visual objects, as depicted in Fig. 1. While LLMs use pre-specified tokenizers [22, 23, 24], the compositional granularity for *implicit* visual understanding is unspecified. On the other hand, slot methods induce object-level representations but fail to learn composition rules. In this work, we then ask:

*Can analogy-solving enable in-context generalization over object-centric compositional elements of visual entities?*

While many linguistic corpora have been extracted from the Internet for text-based tasks, to the best of our knowledge, no datasets exist for measuring in-context composition capability from visual prompts. To this end, we introduce a suite of three diverse benchmarks in Section 3 built on top of rich combinatorial spaces to systematically test image-based in-context learning abilities: (a) 3D Shapes [25], (b) BitMoji Faces, and (c) CLEVr Objects [26]. If the attention over word tokens enables analogical understanding in LLMs, can we transfer this mechanism to visual analogies? We formulate a unifying framework in Section 4.1 to formalize this notion. We present a visual analogy-based meta-learning algorithm called *Im-Promptu* in Section 4.2 and use it to train agents with a cross-attention module. How does one tokenize visual inputs? To answer this, we model several generative agents in Section 5, encompassing a continuum ranging from strong compositional learners using object-centric abstractions to non-compositional monolithic representation learners. Extensive experiments presented in Section 6 show the dependence of robust in-context learning for visual stimuli on object-centric slots as visual tokenizers and cross-attention. We demonstrate the use

of our visual analogy framework in task extrapolation (Section 6.2), combinatorial generalization (Section 6.3), and generating counterfactual images (Section 6.5).

## 2 Related Work

In this section, we discuss prior works and draw connections to relevant research areas.

**2.1 Object-Centric Learning:** This growing body of work targets unsupervised decomposition of scenes into a set of object-centric vectors from raw perception data. A common approach involves modeling an autoencoder consisting of a latent factorization of object slots that are independently decoded to reconstruct the scene [4, 5, 6, 27, 28, 29]. These methods effectively segment scenes into representations of underlying objects but do not learn the flexible combination of object representations. On the other hand, the authors in [7] replace the independent slot decoding bias with an Image-GPT [30] and demonstrate novel slot composition. While this work largely answers how to compose a *given* set of visual entities, their slot selection is performed manually through organization of slots into clustered libraries. In our work, we use a meta-learning approach to help the learner implicitly understand the composition task. Recently, the authors of [31] recast the iterative slot refinement problem as finding of fixed points through implicit differentiation to improve the stability and computational cost of the slot-attention (SA) technique. The authors of [32] use specific inductive biases to disentangle object contents from spatial information that enables relational reasoning. Our work produces a more generalized composition beyond spatial relations.

**2.2 Analogical Reasoning:** The ability to make analogies has been shown to have important implications in various domains, such as problem-solving, creativity, learning, and counterfactual thinking [1, 33, 34, 35, 36]. Various models have been proposed to explain the underlying mechanisms of analogy-making. Popular symbolic approaches include ARGUS [37], Transfer Frames [38], and Structural Mapping Engine [39], connectionist models include LISA [40] and Star-1 [41], and models that occupy a middle ground like Copycat [42]. Contemporary deep-learning models [43, 44, 45] attempt to solve Raven's Progressive Matrices and Bongard Problems, types of visual analogies present in IQ tests. The authors of [46] propose a generative deep visual analogy-making model based on a parallelogram regularizer on the latent space to disentangle variation factors and traverse transformation manifolds. Most similar to us, a recent work [47] reformulates the analogy-making task as Inpainting in an image grid and proposes an auto-encoder pre-training task to that end.

**2.3 Compositional Generative Models:** Prior works [48, 49] use energy-based models to represent composable factors of variation of a scene. However, these methods use concept labels or are grounded in text and are limited to a few concept composition operations. A follow-up work [50] describes unsupervised energy-based concepts; however, the re-composition is performed via manually-picked concepts. Recent works in text-to-image models use language grounding to induce compositionality. DALL-E [9] uses language modeling over the joint space of the text and image tokens to facilitate image generation. The latest text-to-image models [10, 11, 12] use diffusion modeling to guide the image generated from the text. A growing body of work [51, 52, 53] explores text-grounded controllable image editing via prompt engineering, where most parts of the image are preserved, but specific objects are re-composed.

**2.4 Modular Representations and Attention:** A swath of recent works proposes neural methods that combine modular primitives with recurrent mechanisms and cross-attention. The authors of [54] propose the Recurrent Independent Mechanisms (RIMs) to learn weakly coupled dynamical systems with independent compositional neural modules that communicate via a sparse attention bottleneck. Another work [55] uses object-centric abstractions as 'files' to store factorized declarative knowledge and the dynamics of the environment. A follow-up work [56] introduces neural abstractions of the classical Production Systems framework to learn rule-based templates, thereby facilitating compositional dynamics of the environment that can quickly adapt to novel stimuli. Most recently, the authors in [57] propose an attention-based architecture to learn in-context causal relationships between entities.

## 3 Benchmarks

Humans can contextualize past visual stimuli in the face of everyday experiences and even use them for imagining counterfactuals. Having seen an oak table, we can, with little difficulty, look at a metal armchair and understand the concept of an oak armchair. How do we then systematically test such

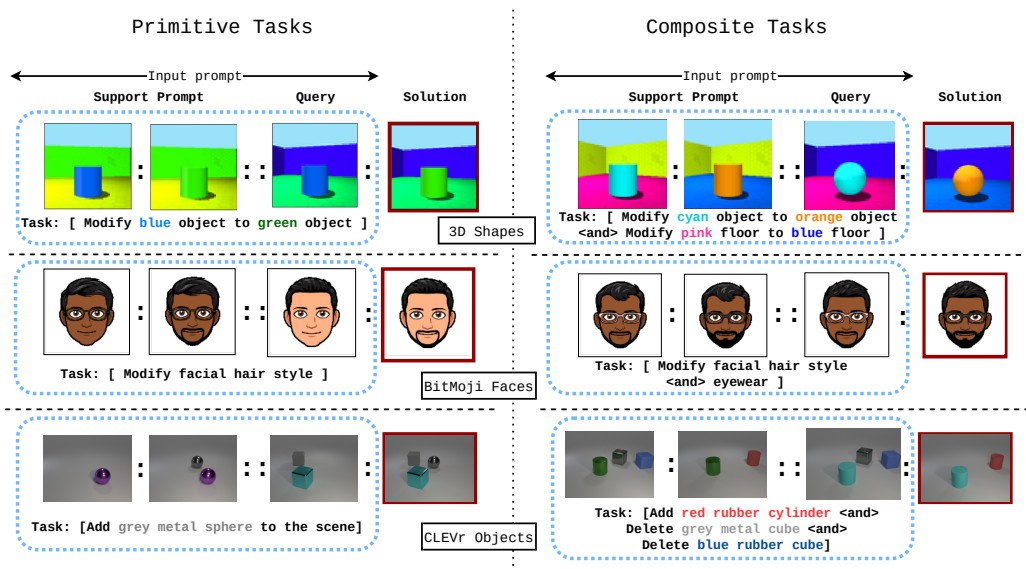

Figure 2: We use combinatorial image spaces to set up visual in-context learning tasks. Each task requires the generation of a solution from a query and supporting examples. The latent description of the task is described within the square parentheses. Left: Primitive tasks modify a composable element in isolation. Right: Composite tasks combine primitives to test the combinatorial generalization skills of the learner.

abilities of a visual in-context learner? In this work, we posit that an answer to this question lies in combinatorial visual spaces with controllable factors of variation, which provide a *semantically rich, simple, scalable, and composable* avenue for setting up such tasks. We create a suite of three benchmarks (Fig. 2) from compositional image creators that include (a) 3D Shapes [25], (b) BitMoji Faces, and (c) CLEVr Objects [26]. Each benchmark has a split of primitive training tasks and out-of-distribution test tasks. The *primitive* tasks (Fig. 2, left) constitute an analogy where a *single* composable element of the visual stimulus is modified in isolation. The description of each benchmark with its primitive set of tasks is given next:

**3.1 3D Shapes Image Prompts:** This dataset consists of static scenes of various objects lying on a colored floor in front of a colored wall viewed from different orientations. To set up primitive tasks, we consider the four properties $P = \{$`object color`, `wall color`, `floor color`, `scene orientation`$\}$, where each property can take multiple values within a domain. For each such task, the latent relation $r(.)$ for the analogy prompt takes a property $p \in P$ and modifies it from a *source* domain value to a *target* value. This benchmark comprises 80000 such tasks with a maximum of four supporting image pairs for each instance and with a roughly equal split across properties.

**3.2 BitMoji Faces Image Prompts:** BitMoji is an avatar creator service for social media users that allows them to create intricate cartoon faces. We queried the BitMoji API [58] to collect faces with four underlying dynamic elements $P = \{$`skin tone`, `facial hair style`, `hair style`, `eyewear`$\}$. Much like the previous case, primitive tasks are generated by intervening on a *source* value of a property and modifying it to a *target* value. We populate the training set with 80000 tasks, with four demonstrations available per task.

**3.3 CLEVr Objects Image Prompts:** CLEVr is a popular visual question-answering dataset with the visual component consisting of multiple objects lying in a scene. We use the CLEVr rendering engine to set up primitive tasks that include `adding` and `deleting` the same object across various scenes. In this fashion, we generate 60000 tasks with three examples per task.

A composable image space allows the concurrent modification of individual visual elements. One can construct *composite* tasks (Fig. 2, right) out of a combination of primitives shown in the training set. A $k$-composite task is denoted as $R^k(.) = r_1 \circ r_2 \circ \cdots \circ r_k$ where $r_1, \cdots, r_k \in \mathcal{R}$. We provide details of the $k$-compositeness test of each of our agents in Section 6.3. Finer details about each

benchmark, along with visual examples from every task, are detailed in Appendix A. We will release the complete dataset of primitive and composite tasks upon the publication of this article.

## 4 Methods

To learn compositional structure over visual stimuli, we begin by formulating a unified mechanistic interpretation of an in-context compositional learner using insights from design elements of LLMs.

**4.1 In-Context Learning as Analogy Completion:** A general *compositional* in-context learner $\mathcal{M}_{\phi,\alpha,\theta}(.)$ can be formalized via the following three key components:

- An encoder $\mathcal{E}_\phi(.)$ that maps the input space $\mathcal{X}$ to the compositional space $\mathcal{C} \triangleq \mathcal{E}_\phi(.) : \mathcal{X} \longmapsto \mathcal{C}$
- An executor $\mathcal{T}_\alpha(.,.,.)$ that maps compositional entities from example tasks, example solutions, and a query task to the query solution $\triangleq \mathcal{T}_\alpha(.,.,.) : \mathcal{C} \times \mathcal{C} \times \mathcal{C} \longmapsto \mathcal{C}$
- A decoder $\mathcal{D}_\theta(.)$ that maps the compositional space back to the input space $\triangleq \mathcal{D}_\theta(.) : \mathcal{C} \longmapsto \mathcal{X}$

Coupling the above functions together yields the learner:

$$\mathcal{M} \triangleq \mathcal{D}_\theta\left(\mathcal{T}_\alpha\left(\mathcal{E}_\phi(.), \mathcal{E}_\phi(.), \mathcal{E}_\phi(.)\right)\right) : \mathcal{X} \times \mathcal{X} \times \mathcal{X} \longmapsto \mathcal{X} \tag{1}$$

The inclusion of $\mathcal{E}_\phi(.)$ and $\mathcal{D}_\theta(.)$ is crucial for compositional abstraction since the input space may not be *a priori* composable as in the case of images. Beyond this, we do not place any strong parametric constraints on the model. Let $\mathcal{R}$ denote a task set. For any task $r \in \mathcal{R}$, define the task analogy $\zeta$ over input pairs $x_1, x_2$ as $x_1 : r(x_1) :: x_2 : r(x_2) \triangleq A : B :: C : D$. An effective in-context learner over $\mathcal{R}$ then satisfies the following property [59]:

$$\forall r \in \mathcal{R}, \ \mathbb{E}_{\zeta \sim \mathcal{P}(r,\mathcal{X})}\left[\mathcal{L}\left(D; \mathcal{M}_{\phi,\alpha,\theta}(A, B, C)\right)\right] \leq \epsilon \tag{2}$$

Here, $\mathcal{P}$ denotes the distribution of the analogies resulting from the input space $\mathcal{X}$ and task $r(.)$, $\mathcal{L}(.)$ denotes a loss metric, and $\epsilon$ is an error bound. The above property simply states that a good in-context learner solves analogies defined over a task set within some finite error bound. In Appendix B, we describe LLMs as an instantiation of this framework.

**4.2 Im-Promptu Learning:** Pre-training LLMs on large-scale text data leads to the emergence of highly transferable internal representations that allow them to adjust to novel tasks by simple priming on a few demonstrations. However, it is still unclear whether the underlying structure of the training data, architecture, or inductive biases instilled by choice of pre-training learning algorithms cause these intriguing properties to emerge. How can we encourage the emergence of such a general-purpose representation for visual data? In this work, we take an explicit approach based on the formalism presented in the previous section. Having established the desideratum (Eq. 2), we set up a learning procedure called *Im-Promptu* that trains the model to generate analogy completions from image prompts. To this end, given a task set $\mathcal{R}$, in each minibatch, we sample an analogy $A : B :: C : D$ that follows a latent primitive $r(.) \in \mathcal{R}$. Given a loss criterion, $\mathcal{L}$, we make the following stochastic gradient updates of the model parameters:

$$\{\phi^{t+1}, \alpha^{t+1}, \theta^{t+1}\} = \{\phi^t, \alpha^t, \theta^t\} - \frac{\alpha}{N}\sum_{i=1}^{N} \nabla_{\theta^t,\alpha^t,\phi^t}\mathcal{L}\left(D, \mathcal{M}_{\phi^t,\alpha^t,\theta^t}(A, B, C)\right) \tag{3}$$

where the step size $\alpha$ is a hyperparameter. The full *Im-Promptu* algorithm is laid out in Algorithm 1.

## 5 Learning Agents

In this section, we explore various modeling choices for a visual in-context learner ranging from simple pixel-space rules to object-centric learners (OCLs).

**5.1 Pixel Baseline:** This non-parametric baseline manipulates the pixel space using a simple addition rule. We denote the support prompt as $A : B$, the query as $C$, and the solution as $\hat{D}$. Then the Pixel baseline predicts $\hat{D}$ as the outcome of a linear transformation, $\hat{D} = C + (B - A)$.

**5.2 Monolithic Learner:** Our first learning agent, inspired by [46], is entirely non-compositional and uses latent monolithic vectors to execute the image prompts as follows:

$$\mathcal{V} = f_\alpha\left([e_\phi(A), e_\phi(B)]\right), \ \hat{D} = d_\theta\left(h_\beta\left([\mathcal{V}, e_\phi(C)]\right)\right) \tag{4}$$

---

**Algorithm 1** Im-Promptu Learning Algorithm

---

**Require:** Task set $\mathcal{R}$, Step Size $\alpha$
Initialize parameters $\theta, \alpha, \phi$ of model $\mathcal{M}$
**while** not done **do**
    Sample primitive task $r(.) \sim \mathcal{R}$
    Sample $N$ input image pairs $X_1^{1:N}, X_2^{1:N} \sim \mathcal{P}_{train}$
    **for** $i$ in $1 \cdots N$ **do**
        Sample analogy $\zeta^i \triangleq x_1^i : r(x_1^i) :: x_2^i : r(x_2^i) \triangleq A : B :: C : D$
    **end for**
    Update $\{\theta, \alpha, \phi\} = \{\theta, \alpha, \phi\} - \frac{\alpha}{N} \nabla_{\theta, \alpha, \phi} \sum_i \mathcal{L}_{\mathcal{M}}(\zeta^i)$
**end while**

---

The encoder $e_\phi(.)$ and decoder $d_\theta(.)$ are convolutional and deconvolutional networks, respectively. The inference network $f_\alpha(.)$ and executor network $h_\beta(.)$ are modeled as multi-layered perceptrons. Here, $[\,]$ denotes the concatenation operator. This architecture is detailed in Appendix D.1.

**5.3 Inpainting Model:** Inspired by recent work on inpainting for In-Context learning [47], this agent (Inpaint., visualized in Appendix D.2) is modeled via an autoencoder architecture that fills in missing patches of an image. The architecture is composed of a discrete variational autoencoder (dVAE) that discretizes the input patches and a Vision Transformer [60] auto-encoder to impute missing values. While the monolithic agent separately encodes the components of the visual analogy, this model jointly represents them as a $2 \times 2$ image grid. The model is pre-trained using the masked autoencoder reconstruction task [61]. Subsequently, at inference time, the analogy solution is represented as a missing image in the grid. Details have been given in Appendix D.2.

**5.4 Object-Centric Learner (OCL):** This agent, as depicted in Fig. 3, is a strongly compositional learner that uses object-centric inductive biases. The encoder and decoder of the framework in Section 4.1 are parametrized using Slot Attention Transformer (SLATE) [7] (see Appendix C.2) as follows:

$$\mathcal{E}_\phi(.) \triangleq SA_\phi(f_\phi^{\text{dVAE}}(.)), \ \mathcal{D}_\theta(.) \triangleq g_\theta^{dVAE}(g_\theta^{GPT}(.)) \tag{5}$$

A Context Encoder Transformer (CET) $\triangleq \mathcal{T}_\alpha(.\,,.\,,.)(.)$ with interleaved layers of: $(a)$ cross-attention on the context slots and $(b)$ self-attention, induces sparse modifications on the query slots to complete the analogy.

$$S_{1:N}^A = \mathcal{E}_\phi(A), \ S_{1:N}^B = \mathcal{E}_\phi(B), \ S_{1:N}^C = \mathcal{E}_\phi(C) \tag{6}$$

$$S_{1:N}^D = CET\left(\text{query} = S_{1:N}^C, \text{keys, values} = S_{1:N}^A, S_{1:N}^B\right) \tag{7}$$

The latent sequence $\hat{Z}_D$ predicted by Image-GPT is mapped by the dVAE back to the pixel space $\hat{D}$.

$$\hat{Z}_D = g_\theta^{GPT}(S_{1:N}^D), \ \hat{D} = g_\theta^{dVAE}(\hat{Z}_D) \tag{8}$$

The CET forward pass is shown in Appendix D.3.

**5.5 Sequential Prompter:** This (Seq., visualized in Appendix D.4) is an ablation over the OCL that replaces CET with an LLM-like sequence prompt obtained from a simple concatenation of slots of images $A$, $B$, and $C$, injected with position embeddings $p_i$.

$$\mathcal{T}_\alpha(.,.,.) \triangleq [S_{1:N}^A, S_{1:N}^B, S_{1:N}^C] + p_i \tag{9}$$

While the OCL explicitly encodes the entire context into a fixed-length slot sequence via the CET, the ablated agent models a longer concatenated sequence.

**5.6 Patch Learner:** This agent straddles the compositionality extremes of Monolithic Learners and OCLs, and uses image patch abstractions. The SA module from OCL is ablated to get a discrete sequence of $4 \times 4$ patch latents as the degree of compositional granularity $\mathcal{E}_\phi(.) \triangleq f_\phi^{\text{dVAE}}(.)$. A larger image implies longer sequences for this modeling choice.

# 6 Experiments

In this section, we set up experiments to answer (1) whether analogical reasoning enables in-context generalization, (2) does such a generalization require key mechanisms from LLMs, namely compositionality and attention, (3) if compositionality is a key ingredient, what is the correct granularity for visual tokens, and (4) what are the limits to the generalization.

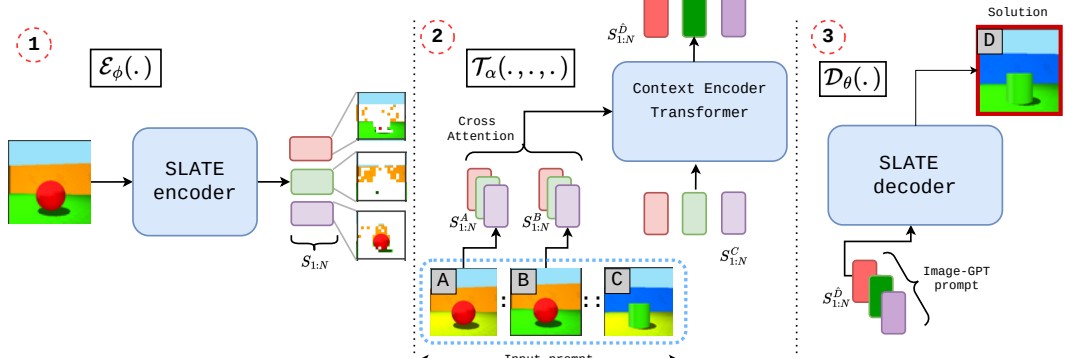

Figure 3: The OCL uses compositional object vectors to answer image prompts. (**1**) The encoder $\mathcal{E}_\phi(.)$ is a pre-trained SLATE [7] encoder that uses a dVAE to obtain a latent discrete sequence for the input image and runs SA [6] over the sequence to extract object-centric slots. (**2**) The CET modifies query slots $S_{1:N}^C$ via cross-attention over the support prompt slots $S_{1:N}^A, S_{1:N}^B$. (**3**) The modified slots are used to prompt an Image-GPT [30] that outputs the latent discrete sequence of the task solution $D$ that is decoded by the dVAE decoder.

**6.1 Training Setup:** We trained each agent on the primitive set of tasks across the three benchmarks using *Im-Promptu* (Section 4.2). The SA-based object-centric encoders used in the OCL and Sequential Prompter were pre-trained as a SLATE autoencoder, as described by the authors in [7]. On the other hand, the Patch Learner was trained completely from scratch. The loss metric $\mathcal{L}$ used to train the agents was cross-entropy (CE) loss between the true latent sequence $Z_D$ and the predicted solution $\hat{Z}_D$ obtained from Image-GPT, i.e., $\mathcal{L}_{impromptu} = CE(Z_D, \hat{Z}_D)$.

In addition to the above loss, the dVAE was trained using the mean-squared error (MSE) loss over the raw pixel space to yield the full loss function $\mathcal{L} = \mathcal{L}_{impromptu} + MSE(D, \hat{D})$. For inference, answers of the transformer-based agents were sampled from the Image-GPT decoder using top-$k$ nucleus sampling [62]. Hyperparameters for training and inference have been laid out in Appendix E.

**6.2 Primitive Task Extrapolation:** In the first out-of-distribution paradigm, we tested the ability of agents to apply learned rules to different domain pairs. In order to do this, we held out 20% of source-target pairs (see Section 3 for the definition of source and target) from the training primitives and tested the ability of agents to generalize from learned rules to unseen source-target pairs. For example, the `object color` property in the 3D Shapes benchmark can take 10 unique values and $10 \times 9 = 90$ source-target combinations. Thus, an agent was trained on only $90 \times 0.8 = 72$ pairs for object-hue primitives. We made sure that each value in the domain set was shown at least once as either the target or the source.

Fig. 4(a) plots scores of different agents across benchmarks against two key metrics: (1) MSE (lower is better) that quantitatively compares the construction against the ground truth and (2) Fréchet inception distance (FID, lower is better) score to measure the perceptual quality of the composition. We make several interesting observations:

**(R1.1) Simple pixel manipulation produces implausible outputs.** The pixel baseline has poor MSE scores ($\sim 138$ for 3D Shapes, $\sim 895$ for BitMoji) with clearly apparent artifacts in the output. While linear pixel transformations are sufficient when an object is independent of the remaining entities, it is unable to model complex dependencies (see Fig. 4(b),(d) under 'Pixel').

**(R1.2) Inpainting model struggles to generalize beyond i.i.d. inputs.** We observed that the Inpainting model is able to cogently complete analogies on the i.i.d. validation set (see Appendix F.1 for outputs). However, when tested on extrapolated primitives, it only fills in the high-level structure (see Fig. 4 (b), (c)) and, in the case of more complex inputs, produces entirely incoherent outputs (Fig. 4 (d)). We posit that a top-down method like inpainting pre-ordains larger datasets for improved generalization and, as such, suffers from sample inefficiency.

**(R1.3) Monolithic Learners are reasonably effective at generalizing from learned rules on structured stimuli.** These agents have the lowest MSE scores on 3D Shapes and BitMoji Faces, both spatially consistent datasets. However, the output deteriorated over CLEVr Objects where the

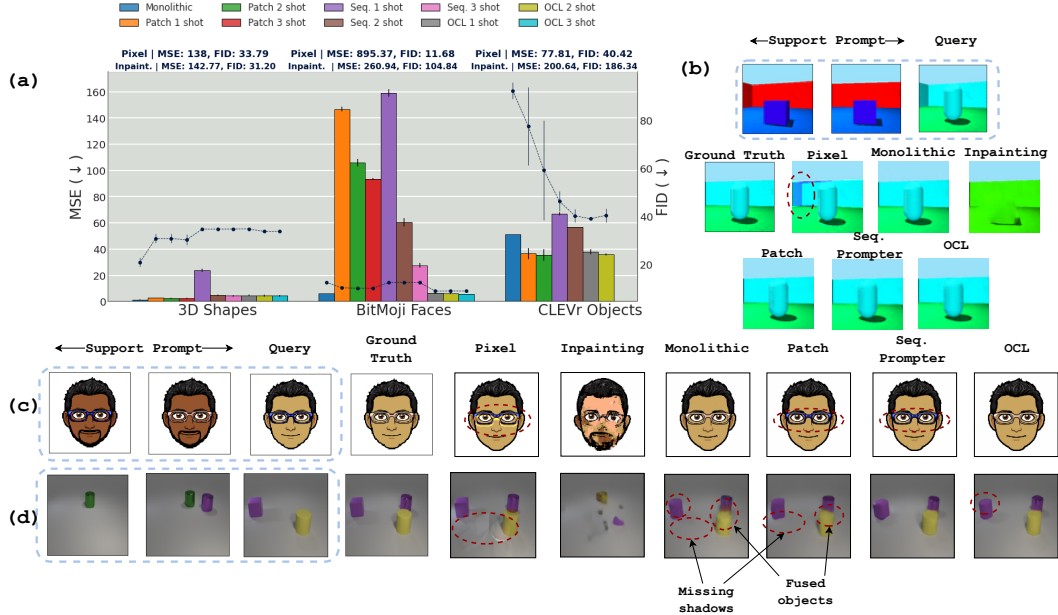

Figure 4: Primitive task extrapolation: (a) Plot showing scores where the left y-axis and the bars represent the MSE scores, while the right y-axis and the dotted line denote the FID scores. (b)-(d) Comparison of agent solutions with the dotted red circle representing anomalies with respect to the ground truth.

latent scene configuration is diverse. The edges of the generated shapes and lighting details were significantly blurred, and occluded objects were often fused together (FID $\sim$ 92, worse than Pixel baseline, Fig. 4(d) under 'Monolithic').

**(R1.4) Patch Learner and Sequential Prompter benefit from additional examples.** For these agents, the addition of context examples was markedly apparent from the significant MSE drops for $> 1$ shot (see Fig. 4(a)). We posit that since these models learn from longer contexts, additional examples are particularly effective in regularizing the output sequence manifold of Image-GPT.

**6.3 Composite Task Extrapolation:** In this generalization paradigm, we probed the compositional generalization capabilities of the agents. A $k$-composite task ($k \geq 2$, see Section 3) is an unordered composite function of $k$ primitives. The value of $k$ is varied across each benchmark, where higher values of $k$ yield visually complex contexts. To solve such a task, the agent must identify the underlying primitive relations and combinatorially modify the query.

In Fig. 5(a), we note (R1.1) and (R1.3) from the previous setup in this extrapolation paradigm as well, but we make additional key observations next:

**(R2.1) The effect of object-centric biases is strong.** This is indicated by the widening MSE gap between the OCL and monolithic agent for increasing values of $k$ across all three benchmarks (see Fig. 5(a)-(c)). Moreover, the OCL and the Sequential Prompter generate outputs with consistent perceptual quality (lower FID scores).

**(R2.2) Monolithic and Patch Learners apply shortcuts.** The monolithic agent tends to produce a Frankenstein-like image, as if a superposition of context images (see Fig. 5(d)-(f) under 'Monolithic'). In the case of the CLEVr Objects dataset, both the Monolithic and Patch Learners often simply filled in 'blobs' agnostic to the color and shape of objects (Fig. 5(f), Appendix F.4), with missing shadows leading to lower MSE scores but poor visual quality.

**6.4 Analysis:** Next, we discuss key elements required for in-context composition.

**(A1) In-context generalization critically hinges on compositional granularity.** As evident from (R1.2), monolithic representations suffice for reliably learning primitive composition rules over simple contexts. However, they suffer from fidelity issues when the visual space becomes more complex. Beyond primitive generalization, patch abstractions are effective when the underlying space

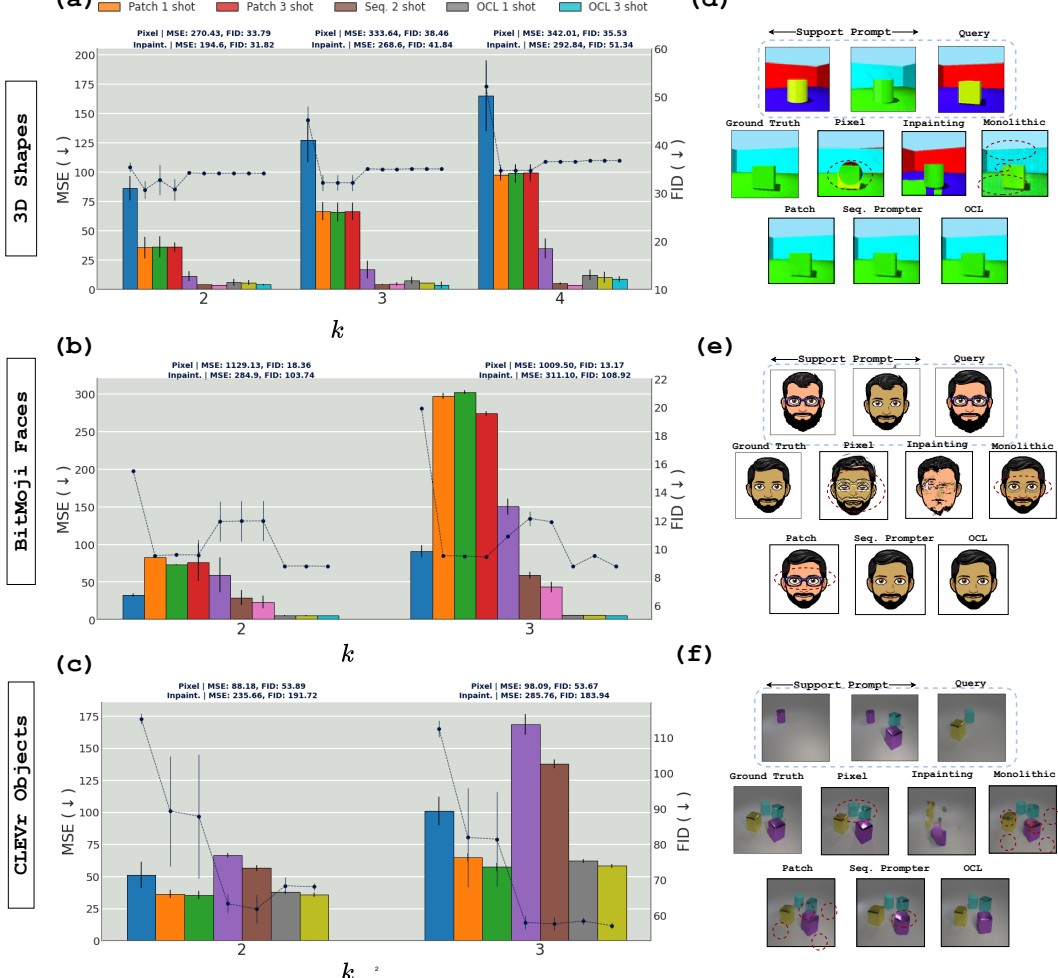

Figure 5: Composite task extrapolation: (a)-(c) Plots showing scores across benchmarks where the left y-axis and the bars represent the MSE scores, while the right y-axis and the dotted line denote the FID scores. $k$ denotes the level of compositeness. (d)-(f) Comparison of agent solutions with the dotted red circle representing anomalies with respect to the ground truth.

has object-patch equivalence, as evident from the CLEVr Objects dataset (Fig. 5(c)). However, even with object-patch equivalence, the Patch Learner omits the inter-object dependencies (e.g., object occlusions and shadows). It further fails to learn complex and diverse objects spanning multiple patches reliably (does poorly on the BitMoji Faces dataset with diverse facial components). Slot-based compositionality with the CET not only enables the generalization of the learned primitives but also enables combinatorial generalization to composite tasks that require abstracting out the composition rule over each entity in isolation as well as composing the resultant interplay between them to generate a globally consistent output.

**(A2) Encoding the context via cross-attention leads to sample efficiency.** Longer slot sequences in the Sequential Prompter require multi-shot contexts across all benchmarks. The sample inefficiency was particularly observed on the CLEVr Objects dataset that uses six slots and, hence, much longer sequences. Cross-attention via CET is essential for implicit inference, enabling the learner to encode specific objects from the context.

**6.5 'Counterfactual Prompt Engineering' with Im-Promptu:** LLMs have spurred interest in several priming techniques [63, 64] that can 'engineer' the model into following user instructions. While this natural-language-based programming language provides an intuitive interface for tasks that have a fundamental degree of language abstraction (writing code, summarization, symbolic

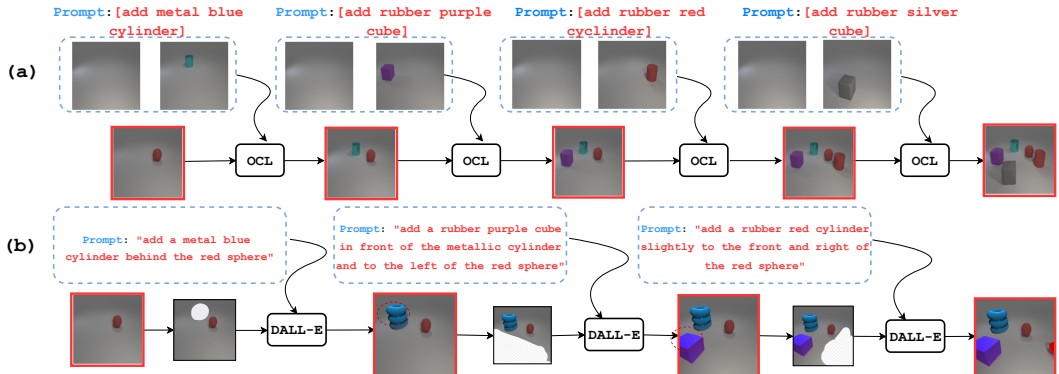

Figure 6: Scene creation from object components: (a) OCL trained via *Im-Promptu* is used to generate a scene of objects. Image-Prompts engineered using existing assets reliably generate the scene object by object. (b) The same scene is generated via language grounding and in-painting features of DALL-E [9]. Object properties are distorted and specifying the location of objects is tedious via language.

manipulation, etc.), it is not *a priori* obvious that textual descriptions should also scaffold image generators. Coming up with natural-language prompts for complex visual entities is often tedious and unintuitive. Instead, explaining an image is naturally easier from its entities and composition over them[1]. To this end, *Im-Promptu* provides an avenue for generating counterfactuals via image prompts. In Fig. 6(a), we demonstrate the creation of a scene from its object components via the OCL. As a point of comparison, in Fig. 6(b), using image-inpainting and natural language prompts with DALL-E [9] yields an unreliable output with distorted objects. More practically, the OCL reduces human effort significantly in rendering images with desired properties by using existing visual assets. The procedure to generate scenes from OCL can be found in Appendix G.

# 7  Conclusion

This work investigated whether analogy-solving can enable in-context compositional generalization over visual entities. We first designed a new test suite. We probed if attention and compositionality, key ingredients in LLMs, also play a crucial role in the visual domain. We transported these ingredients from language to model several agents. Our experiments showed that non-compositional agents do not generalize well beyond primitive task extrapolation. Even with compositionality baked in, the performance of patch-based learners depends on idiosyncrasies of the visual structure. However, we found that object-centric biases consistently facilitate an implicit understanding of composition rules and generate outputs with global semantic consistency. Our ablation of the CET suggested that cross-attention plays a crucial role, just like language models, which posits the importance of analogy and attention in understanding the underpinnings of in-context learning. Future research challenges include collecting real-world primitives from photo-realistic graphic engines or human labels, improving object-centric inductive biases for real-life entities, and designing better prompting techniques. We hope that our work will spur further research on visual in-context learning.

## Acknowledgments and Disclosure of Funding

We would like to thank Sreejan Kumar for helpful discussions and insights throughout the course of this work. The experiments reported in this article were performed on the computational resources managed and supported by Princeton Research Computing at Princeton University. This project was funded by NSF Grant No. CCF-2203399 and ONR Grant No. N00014-18-1-2873. JS gratefully acknowledges financial support from the Schmidt DataX Fund at Princeton University made possible through a major gift from the Schmidt Futures Foundation.

---

[1]"No one is an artist unless he carries his picture in his head before painting it, and is sure of his method and composition" – Claude Monet

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
