# Appendices

## A  Benchmark Details

The full set of primitive tasks is summarized in Table 1 and the statistics detailed in Table 2. Primitives can be individually visualized in Figs. 7-16. Each primitive figure shows an analogy where the underlying relation is over a particular composable property of the scene/image.

| Dataset | Primitive Task Set | Domain Size |
|---|---|---|
| 3D Shapes | (1) Modify object color from source value to target value | 10 |
| | (2) Modify floor color from source value to target value | 10 |
| | (3) Modify wall color from source value to target value | 10 |
| | (4) Modify scene orientation from source value to target value | 15 |
| BitMoji Faces | (1) Modify skintone from source value to target value | 3 |
| | (2) Modify hair style from source value to target value | 10 |
| | (3) Modify facial hair from source value to target value | 5 |
| | (4) Modify eyewear from source value to target value | 5 |
| CLEVr Objects | (1) Add object to the scene at position $p$ | 1000 |
| | (2) Delete object from the scene at position $p$ | 1000 |

Table 1: Summary of primitive tasks across all benchmarks

| Dataset | Examples per task | #primitive tasks | #primitive extrapolation tasks | #$k$-composite tasks |
|---|---|---|---|---|
| 3D Shapes | 3 | 80000 | 1000 | 1000 $\forall k \in \{2, 3, 4\}$ |
| BitMoji Faces | 3 | 80000 | 1000 | 1000 $\forall k \in \{2, 3\}$ |
| CLEVr Objects | 2 | 55000 | 1000 | 200 $\forall k \in \{2, 3, 4\}$ |

Table 2: Benchmark statistics

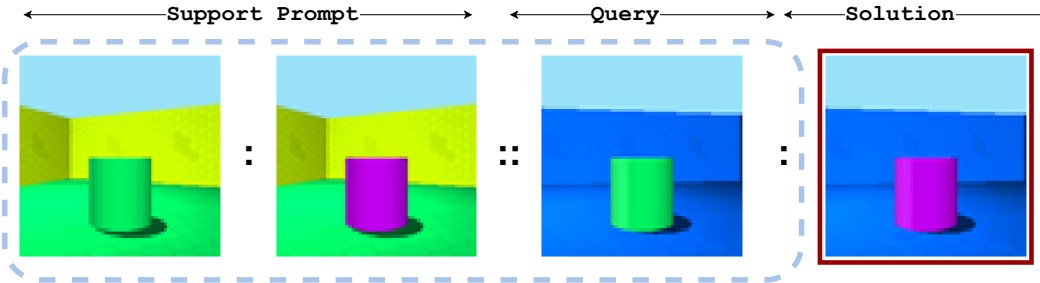

Figure 7: **3D Shapes** → Modify object color from source value to target value.

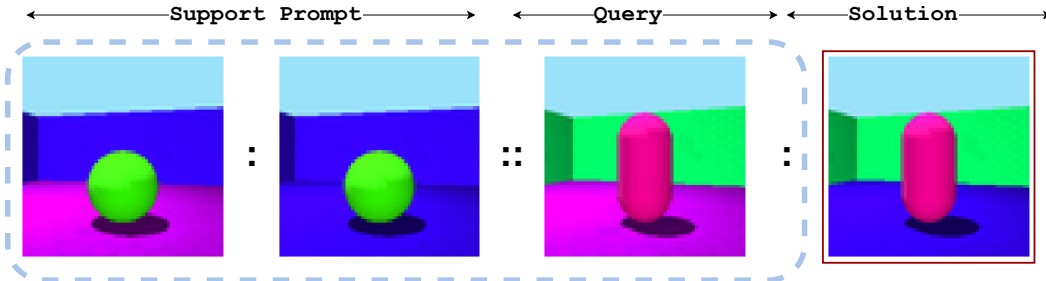

Figure 8: **3D Shapes** → Modify `floor color` from source value to target value.

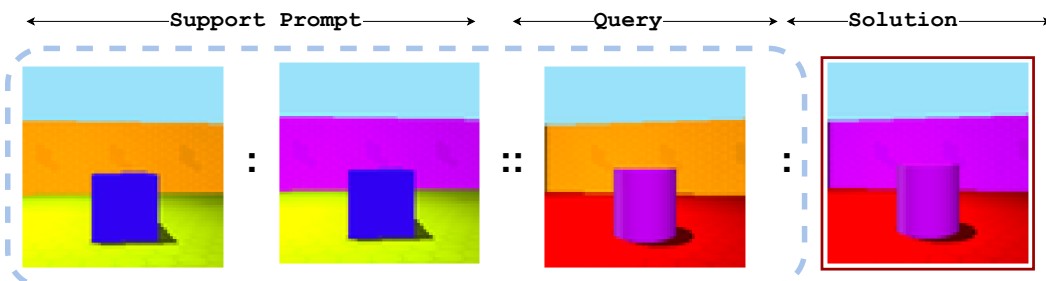

Figure 9: **3D Shapes** → Modify `wall color` from source value to target value.

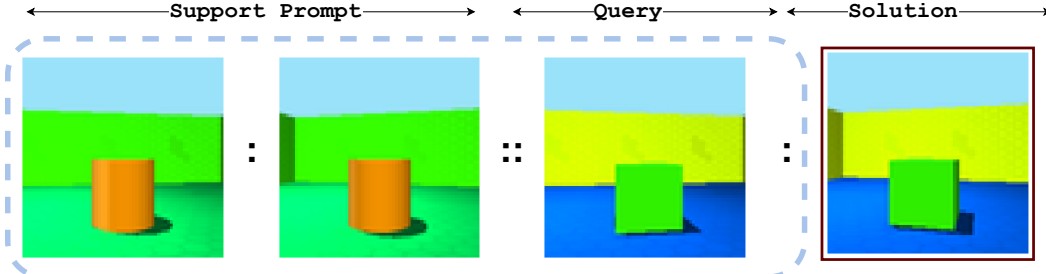

Figure 10: **3D Shapes** → Modify `orientation` from source value to target value.

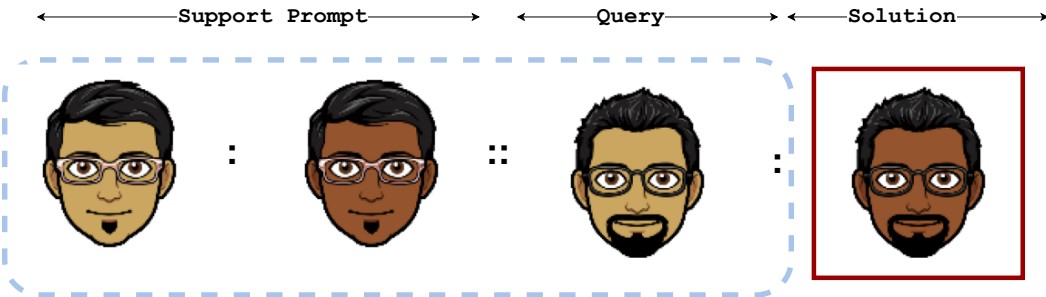

Figure 11: **BitMoji Faces** → Modify `facial skintone` from source value to target value.

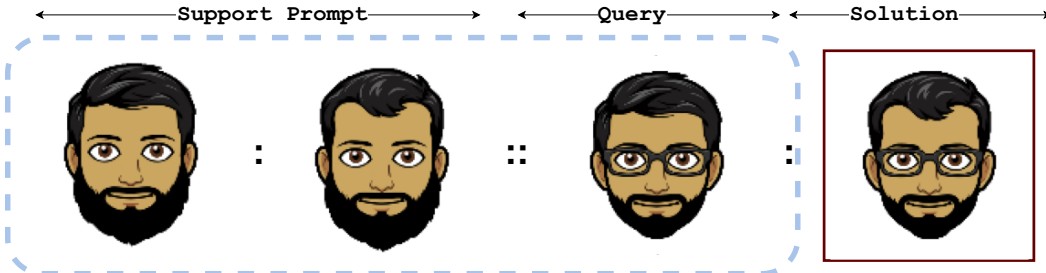

Figure 12: **BitMoji Faces** → Modify `hair style` from source value to target value.

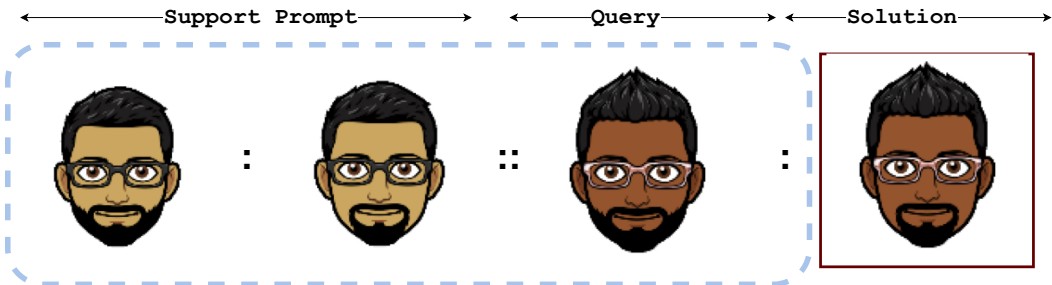

Figure 13: **BitMoji Faces** → Modify `facial hair` from source value to target value.

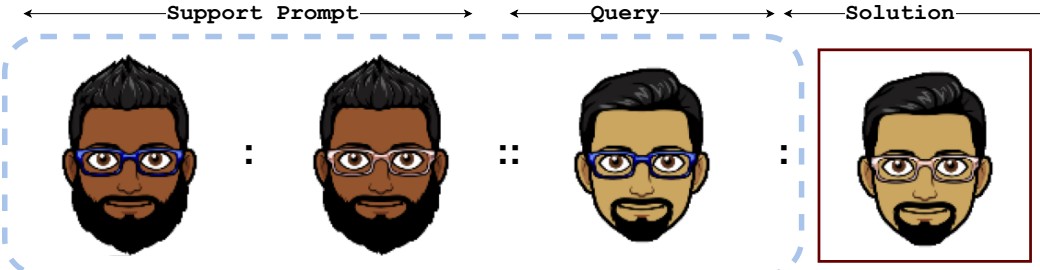

Figure 14: **BitMoji Faces** → Modify `eyewear` from source value to target value.

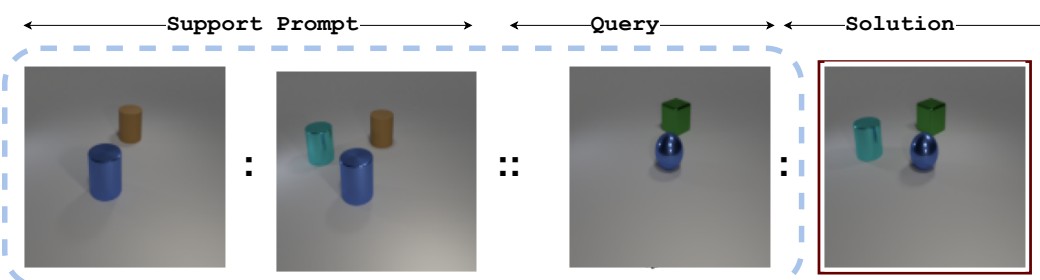

Figure 15: **CLEVr** → Add object to position $p$.

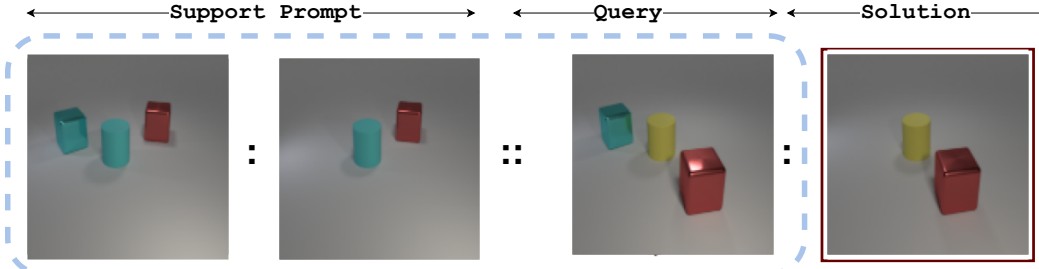

Figure 16: **CLEVr** $\rightarrow$ Delete object from position $p$.

## B   LLMs as In-Context Compositional Learners

It is straightforward to see how the aforementioned framework subsumes LLMs. The input space $\mathcal{X}$ is natural language and the compositional space $\mathcal{C}$ contains the dictionary of resultant tokens obtained via a tokenizer $\mathcal{E}_\phi(.)$. The relation class $\mathcal{R}$ in this context is the set of underlying tasks spanning numerical reasoning, reading comprehension, sentiment analysis, and question answering, to name a few. The core Transformer model forms the executor $\mathcal{T}_\alpha(.,.,.)$ that simply concatenates $A$, $B$, and $C$ into a sequence prompt to compose the output $D$ autoregressively. A de-tokenizer $\mathcal{D}_\theta(.)$ constructs natural language outputs from the generated tokens.

## C   Background

We give brief details of the Slot Attention Mechanism and Slot Attention Transformer (SLATE) that form essential components of our best-performing agents.

**C.1 Slot Attention:** Object-Centric Learning frameworks decompose scenes into compositional object files called slots. Slot Attention (SA) [6] is a powerful auto-encoder-based inductive bias for learning such slots. The encoder $f_\phi(.)$ initializes a set of $N$ symmetric and independent slots $S_{1:N}^0 \triangleq \{s_1^0, s_2^0, \cdots, s_N^0\}$ that iteratively compete over $T$ timesteps to attend to the input scene $X$ and break symmetry to yield the final slots $S_{1:N}^T \triangleq \{s_1^T, s_2^T, \cdots, s_n^T\}$. A decoder $g_\theta(.)$ then individually decodes each slot in the final set into the pixel space to yield $N$ images $I_{1:N}$ and their corresponding masks $M_{1:N}$. A convex combination of the images, using masks as weights, reconstructs the scene $\hat{X}$.

$$S_{1:N}^T = f_\phi(S_{1:N}^0, X) \tag{10}$$

$$I_1, M_1 = g_\theta(s_1^T), \cdots, I_N, M_N = g_\theta(s_N^T). \tag{11}$$

$$\hat{X} = M_1 \times I_1 + \cdots + M_N \times I_N \tag{12}$$

**C.2 Slot Attention Transformer (SLATE):** Recent work [7] solves the pixel independence problem in SA [6] by learning a latent discrete library from pixels using a dVAE [65, 66] and then running SA over the vocabulary space of the dictionary. An Image-GPT decoder [30] learns to predict the sequence of image latent from the slot prompts and, as a result, enables novel composition of slots. We lay out the forward pass of the autoencoder architecture below but defer the finer details to the original article.

$$Z_{1:L} = \{z^1, \cdots, z^L\} = f_\phi^{dVAE}(X) \rightarrow \text{ dVAE encoder} \tag{13}$$

$$S_{1:N}^T = SA_\phi(Z_{1:L}, S_{1:N}^0) \rightarrow \text{Slot Attention} \tag{14}$$

$$\hat{Z}_{1:L} = g_\theta^{GPT}(S_{1:N}^T) \rightarrow \text{Image-GPT} \tag{15}$$

$$\hat{X} = g_\theta^{dVAE}(\hat{Z}_{1:L}) \rightarrow \text{dVAE decoder} \tag{16}$$

Here, $X$ is the input, $Z_{1:L}$ is the latent discrete sequence of length $L$, and $S_{1:N}$ denotes the $N$ slots.

## D   Learning Agents

**D.1 Monolithic:** The encoder $e_\phi(.)$ and decoder $d_\theta(.)$ architectures are convolutional and deconvolutional networks with four layers initialized as shown in Tables 3 and 4, respectively.

| Layer | Stride | Activation | Channels | Weight Matrix |
|---|---|---|---|---|
| Conv $3 \times 3$ | 2 | ReLU | 192 | - |
| Conv $3 \times 3$ | 2 | ReLU | 192 | - |
| Conv $3 \times 3$ | 2 | ReLU | 192 | - |
| Conv $3 \times 3$ | 2 | ReLU | 192 | - |
| Fully Connected | - | ReLU | - | $(192 \times \text{Image Size}/16 \times \text{Image Size}/16) \times 192$ |
| Fully Connected | - | - | - | $192 \times 192$ |

Table 3: Encoder $e_\phi(.)$ of the monolithic agent

| Layer | Stride | Activation | Channels | Weight Matrix |
|---|---|---|---|---|
| Fully Connected | - | - | - | $192 \times 192$ |
| Fully Connected | - | ReLU | - | $192 \times (192 \times \text{Image Size}/16 \times \text{Image Size}/16)$ |
| Conv Transpose $3 \times 3$ | 2 | ReLU | 192 | - |
| Conv Transpose $3 \times 3$ | 2 | ReLU | 192 | - |
| Conv Transpose $3 \times 3$ | 2 | ReLU | 192 | - |
| Conv Transpose $3 \times 3$ | 2 | ReLU | 3 | - |

Table 4: Decoder $d_\theta(.)$ of the monolithic agent

The inference network $f_\alpha(.)$ and executor network $h_\beta(.)$ are feed-forward networks, as parameterized in Table 5.

| Layer | Weight Matrix | Activation |
|---|---|---|
| Fully Connected | $384 \times 192$ | ReLU |
| Fully Connected | $192 \times 192$ | ReLU |
| Fully Connected | $192 \times 192$ | ReLU |

Table 5: The inference network $f_\alpha(.)$ and executor network $h_\beta(.)$ of the monolithic agent

**D.2 Inpainting Model:** The Inpainting model has been described in Fig. 17. The model architecture follows from the task proposed in [47, 61] with the exception of the VQ-GAN replaced by a dVAE and the model trained end-to-end from scratch. Hyperparameters across various test suites have been specified in Table 6.

| Module | Hyperparameters | 3D Shapes | BitMoji Faces | CLEVr Objects |
|---|---|---|---|---|
| Pixel and Discrete Space | Image Size | 64 | 128 | 128 |
| | Image Tokens | 256 | 1024 | 1024 |
| dVAE | Vocabulary Size | 512 | 1024 | 4096 |
| | Tau | 0.1 | 0.1 | 0.1 |
| ViT Encoder/Decoder | Num. Layers | 4 | 8 | 8 |
| | Heads | 4 | 8 | 8 |
| | Hidden Dims | 192 | 192 | 192 |

Table 6: Hyperparameters for the Inpainting model.

**D.3 Object-Centric Learner:** The Context Encoder Transformer (CET) is a stack of $L$ blocks, each computing (a) self-attention over inputs followed by (b) cross-attention of inputs over context slots. The cross-attention implementation of block $l$ is shown in Algorithm 2.

**D.4 Sequential Prompter:** We visualize this learner in Fig. 18.

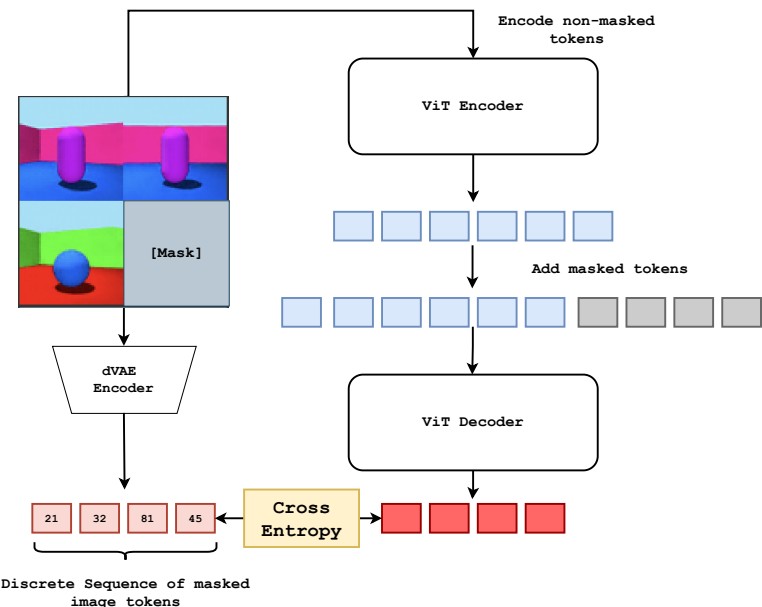

Figure 17: Visualization of the Inpainting model architecture. A dVAE encodes the masked input into a discrete sequence. A ViT [60] encodes the non-masked context patches. Subsequently, masked tokens are added to the encoded patches and decoded via another ViT to predict the discrete sequence. The model is trained end-to-end via the popular Masked Autoencoder (MAE) reconstruction task [61]

---

**Algorithm 2** Cross-Attention for Block $l$ of CET

---

**Require:** $A^{1:N} \in \mathbb{R}^{N \times d}$, $N$ slots of context $A$

**Require:** $B^{1:N} \in \mathbb{R}^{N \times d}$, $N$ slots of context $B$

**Require:** $C_{SA,l}^{1:N} \in \mathbb{R}^{N \times d}$, $N$ slots of input $C$ obtained from the self-attention layer of block $l$

**Output:** $C_l^{1:N}$, Context encoded input slots of block $l$

Get query tokens: $Q^C = MLP_Q(C_{SA,l}^{1:N})$

Get keys, values of $A$, $K^A, V^A = MLP_{KV}(A_{l-1}^{1:N})$

Get keys, values of $B$, $K^B, V^B = MLP_{KV}(B_{l-1}^{1:N})$

Concatenate $K = [K^A, K^B] \in \mathbb{R}^{2N \times d}$
Concatenate $V = [V^A, V^B] \in \mathbb{R}^{2N \times d}$

Compute Cross-Attention Matrix $Att = \text{softmax}(Q^T K / \sqrt{d})$
Compute $C_l^{1:N} = Att \times V$

---

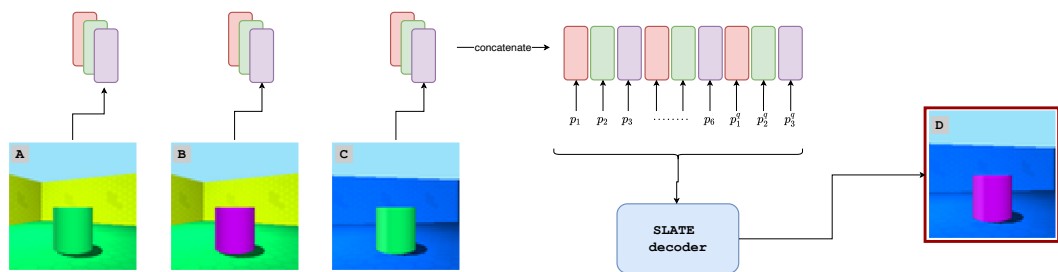

Figure 18: The Sequential Prompter concatenates slots from context prompts and a query together. It further injects position embeddings to create an LLM-like sequential input prompt for the decoder.

# E  Hyperparameters

The hyperparameters for OCL are listed in Table 7. The ablated architectures follow the same setup without the ablated module. All experiments were performed on NVIDIA A100 GPUs.

| Module | Hyperparameters | 3D Shapes | BitMoji Faces | CLEVr Objects |
|---|---|---|---|---|
| Pixel and Discrete Space | Image Size | 64 | 128 | 128 |
| | Image Tokens | 256 | 1024 | 1024 |
| dVAE | Vocabulary Size | 512 | 1024 | 4096 |
| | LR (no warmup) | $5 \times 10^{-5}$ | $5 \times 10^{-5}$ | $1 \times 10^{-5}$ |
| | Tau | 0.1 | 0.1 | 0.1 |
| CET | Num. Layers | 4 | 4 | 4 |
| | Heads | 4 | 4 | 4 |
| | Hidden Dims | 192 | 192 | 192 |
| Image-GPT | Num. Layers | 4 | 8 | 8 |
| | Heads | 4 | 8 | 8 |
| | Hidden Dims | 192 | 192 | 192 |
| Slot Attention | Num. Slots | 3 | 2 | 6 |
| | Iterations | 3 | 3 | 7 |
| | Slot Heads | 1 | 3 | 1 |
| | Hidden Dims | 192 | 192 | 192 |
| | LR (no warmup) | $5 \times 10^{-5}$ | $5 \times 10^{-5}$ | $1 \times 10^{-5}$ |
| Training Setup | Batch Size | 32 | 24 | 20 |
| | LR Warmup steps | 15000 | 30000 | 30000 |
| | Peak LR | $1 \times 10^{-4}$ | $1 \times 10^{-4}$ | $1 \times 10^{-4}$ |
| | Dropout | 0.1 | 0.1 | 0.1 |
| | Gradient Clipping | 1.0 | 1.0 | 1.0 |
| Sampling Scheme | Temperature | 0.7 | 0.7 | 0.1 |
| | Top-$k$ | 8 | 8 | — |
| | Top-$p$ | 0.75 | 0.75 | 0.5 |
| Training Cost | GPU Usage | 15 GB | 40 GB | 40 GB |
| | Days | 1 | 3 | 5 |

Table 7: Hyperparameters for the Object-Centric Learner (OCL) instantiation and training setup

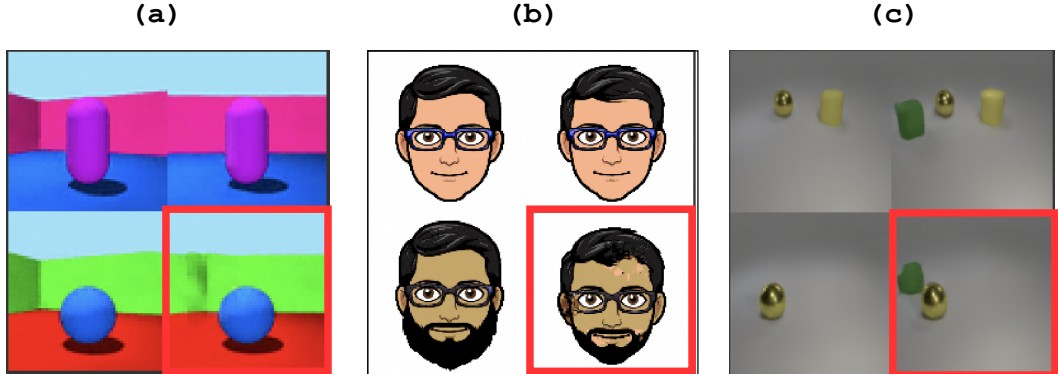

Figure 19: Inpainting model results for the i.i.d. validation set for (a) 3D Shapes (b) BitMoji Faces, and (c) CLEVr Objects analogies. The lower right grid (shown within red boundaries) shows the model completion.

# F    Experiments

**F.1  Inpainting Validation Set Results:** Fig. 19 shows the Inpainting completions for the i.i.d. distribution validation set. The lower right grid is masked and subsequently filled in by the model. While the model produces near-accurate solutions for this set, it is unable to perform coherent out-of-distribution extrapolation.

**F.2  Object Slot Emergence:** Distinct object slots stay preserved in object-centric architectures (OCL and Sequential Prompter) while training on primitive in-context learning tasks. We visualize the slot formation in Figs. 20, 21, and 22 for 3D Shapes, BitMoji Faces, and CLEVr Objects, respectively.

**F.3  Primitive Task Extrapolation:** Figs. 23-25 show generated examples in the primitive task extrapolation regime by different agents.

**F.4  Composite-Task Extrapolation:** Figs. 26-32 show generated examples in the composite tasks extrapolation regime by different agents.

# G    Counterfactual Prompt Engineering

Algorithm 3 details the scene generation and prompt engineering mentioned in Sec. 6.5.

---

**Algorithm 3** Scene Generation via Counterfactual Prompt Engineering

---

    **Require:** $S^{1:T}$, Asset scenes containing individual objects

    **Require** $E$, Empty scene
    **Require** $S^0$, Initial Scene
    **Require:** $M(.)$, OCL Model pre-trained on CLEVr Objects analogies
    **Initialize** Q $\leftarrow S^0$
    **for** $i$ in $1 \cdots T$ **do**
        Set A $\leftarrow S^0$
        Set B $\leftarrow S^i$
        Set C $\leftarrow$ Q
        $\hat{D} = M(A, B, C)$
        Set $Q \leftarrow \hat{D}$
    **end for**
    **Return:** Q

---

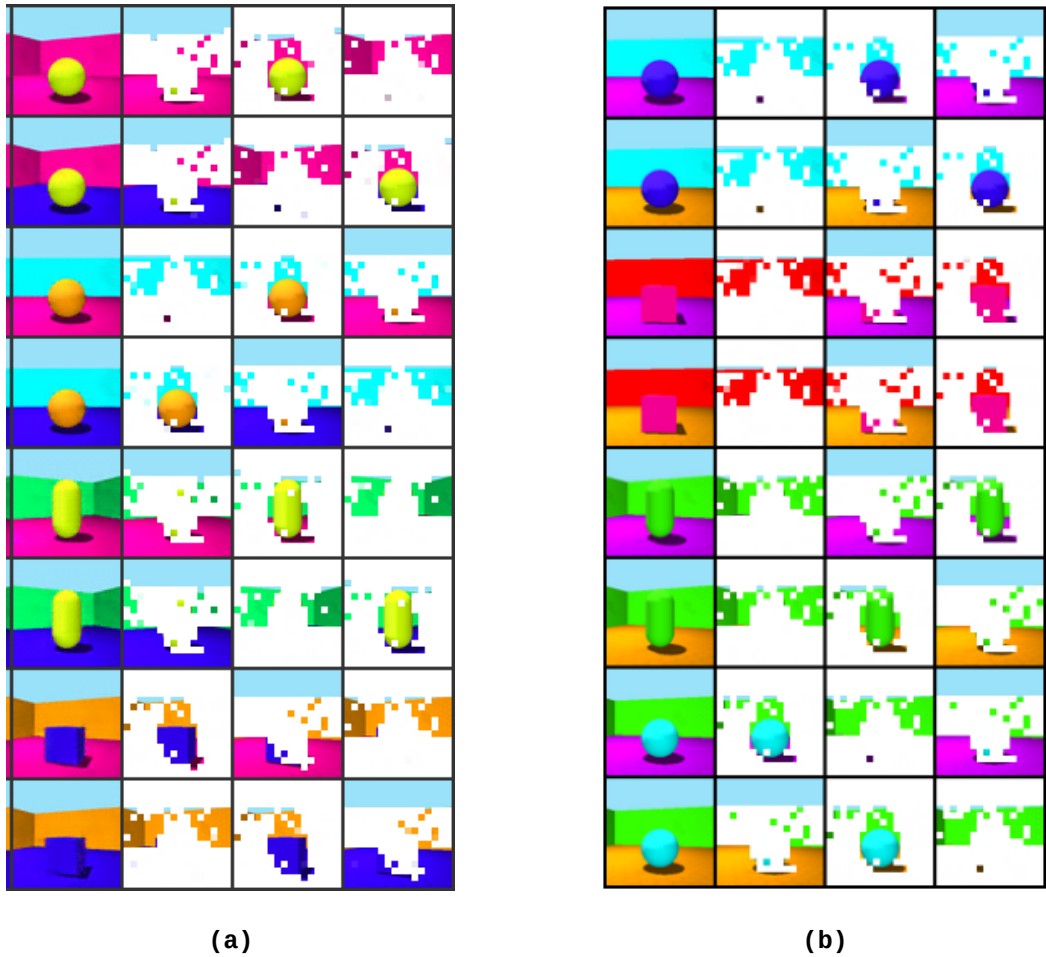

(a)              (b)

Figure 20: Slots from training OCLs on 3D Shapes: (a) pre-trained slots and (b) slots after *Im-Promptu* training. The structure of slots remains preserved.

## Broader Impact

This work demonstrates the potential for analogy-making to enable an in-context understanding of composition rules over visual stimuli. The benchmarks and meta-learning framework presented in this article provide a foundation for further exploration, which could lead to the development of more efficient and effective models for image generation and other visual tasks. While the current benchmark has no sampling bias over primitives, scaling up the models and the dataset can lead to biased outputs and the generation of counterfactual images capable of fooling humans.

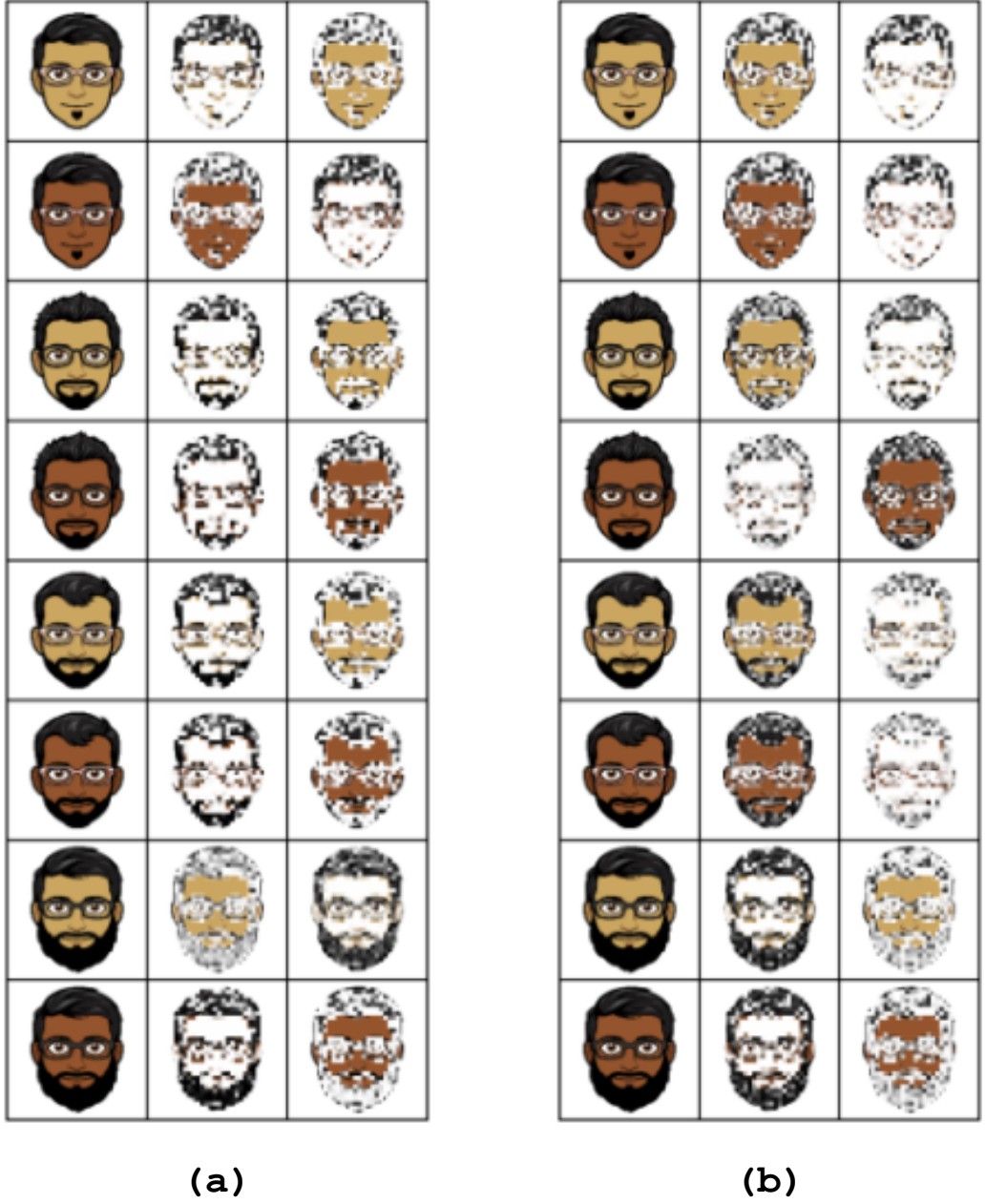

**(a)**                                    **(b)**

Figure 21: Slots from training OCLs on BitMoji Faces: (a) pre-trained slots and (b) slots after *Im-Promptu* training. There occurs a minor modification in slots where the structure of the eye is merged with the slots carrying the facial contents.

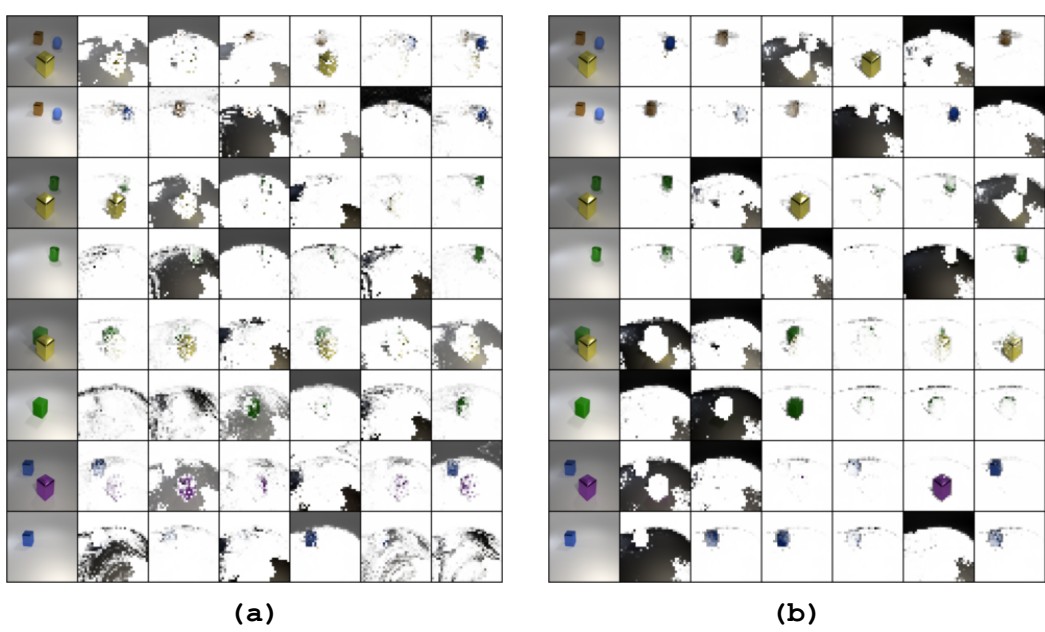

**(a)**                **(b)**

Figure 22: Slots from training OCLs on CLEVr Objects: (a) pre-trained slots and (b) slots after *Im-Promptu* training. We observe that slots become more refined after *Im-Promptu* training.

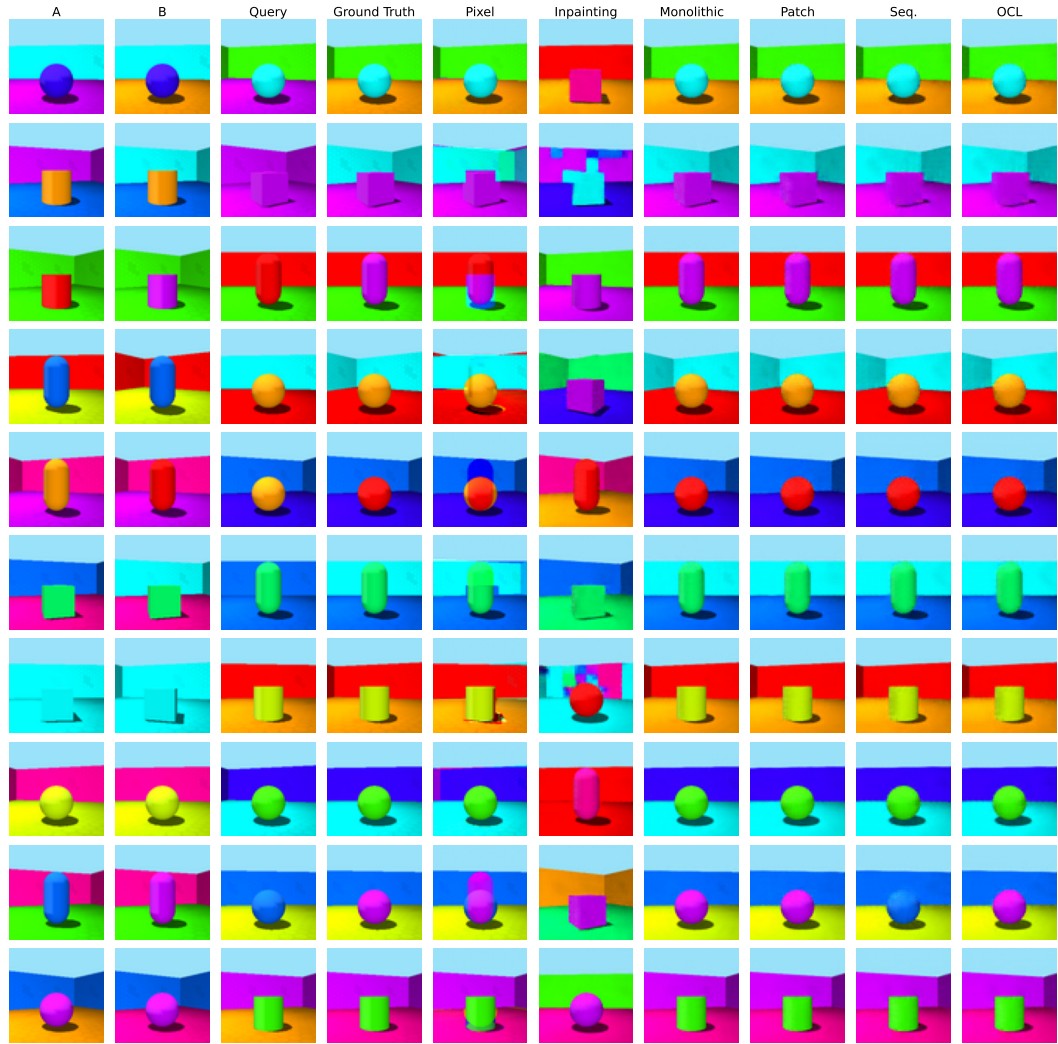

Figure 23: Primitive task extrapolation for the 3D Shapes benchmark.

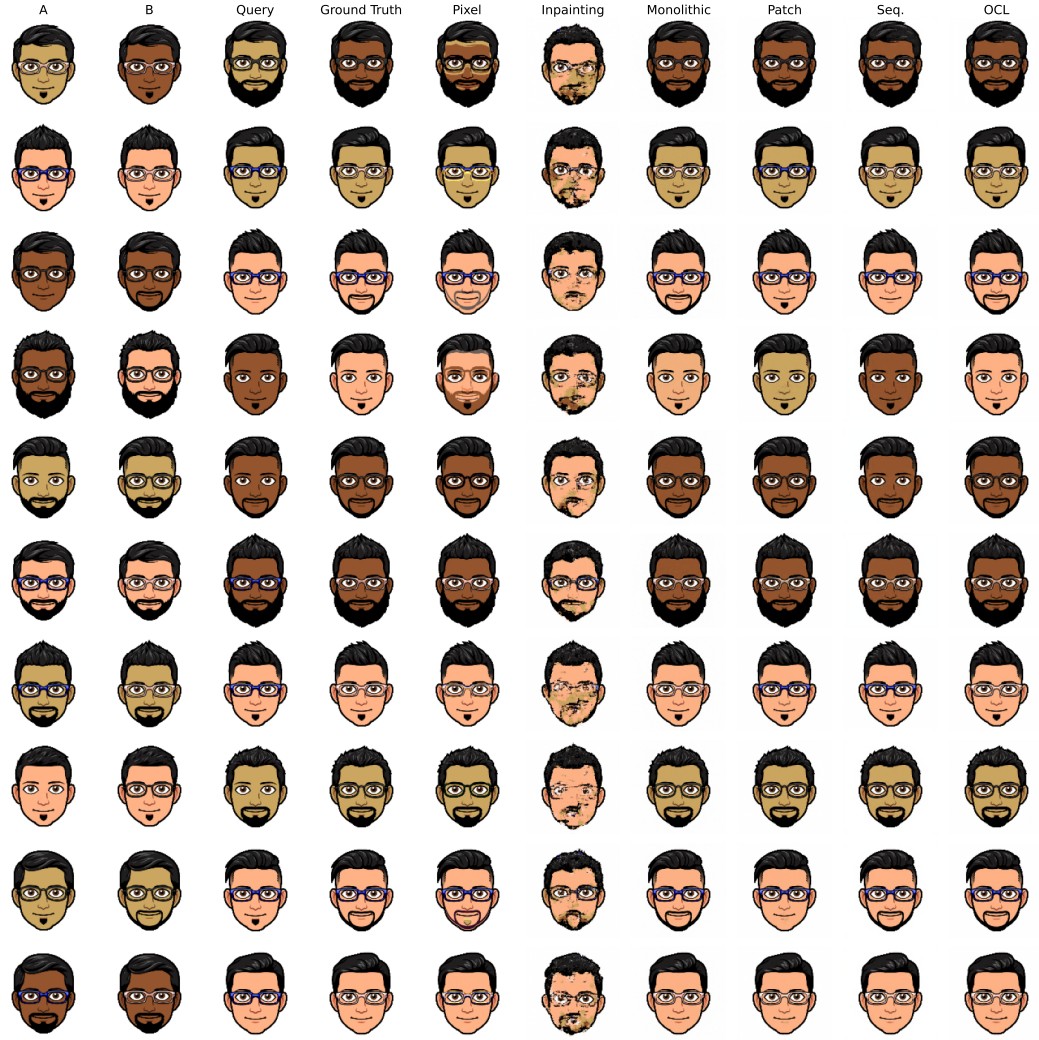

Figure 24: Primitive task extrapolation for the BitMoji Faces benchmark.

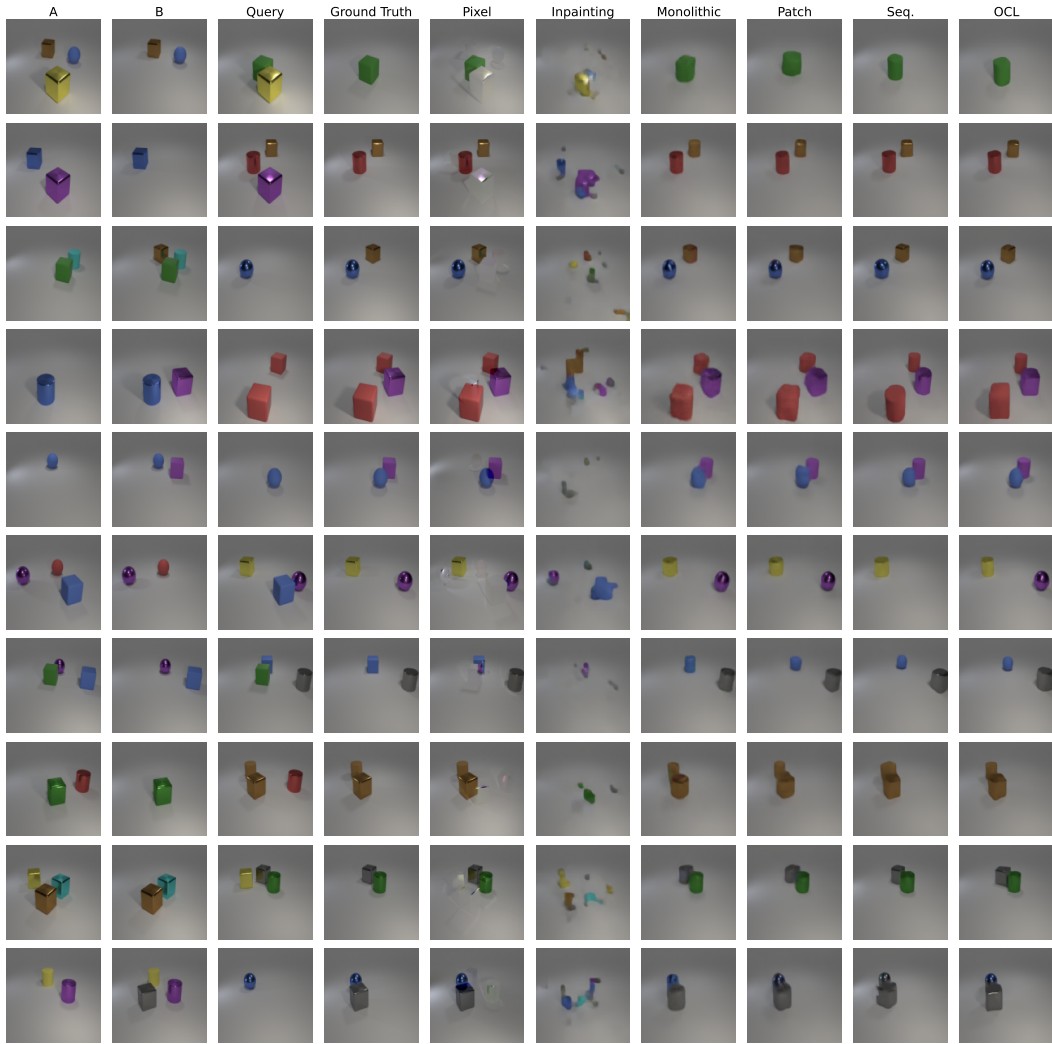

Figure 25: Primitive task extrapolation for the CLEVr Objects benchmark.

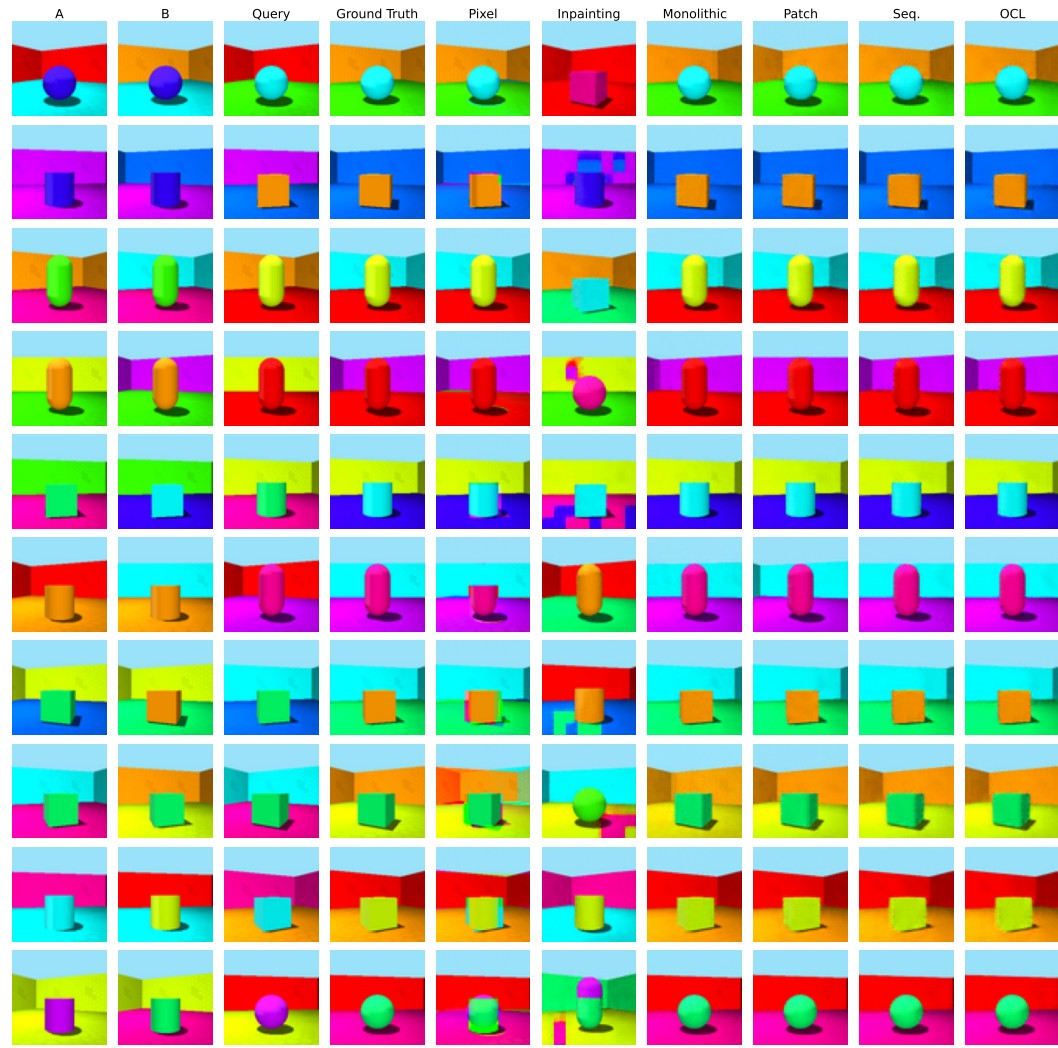

Figure 26: Two-composite task results on the 3D Shapes benchmark.

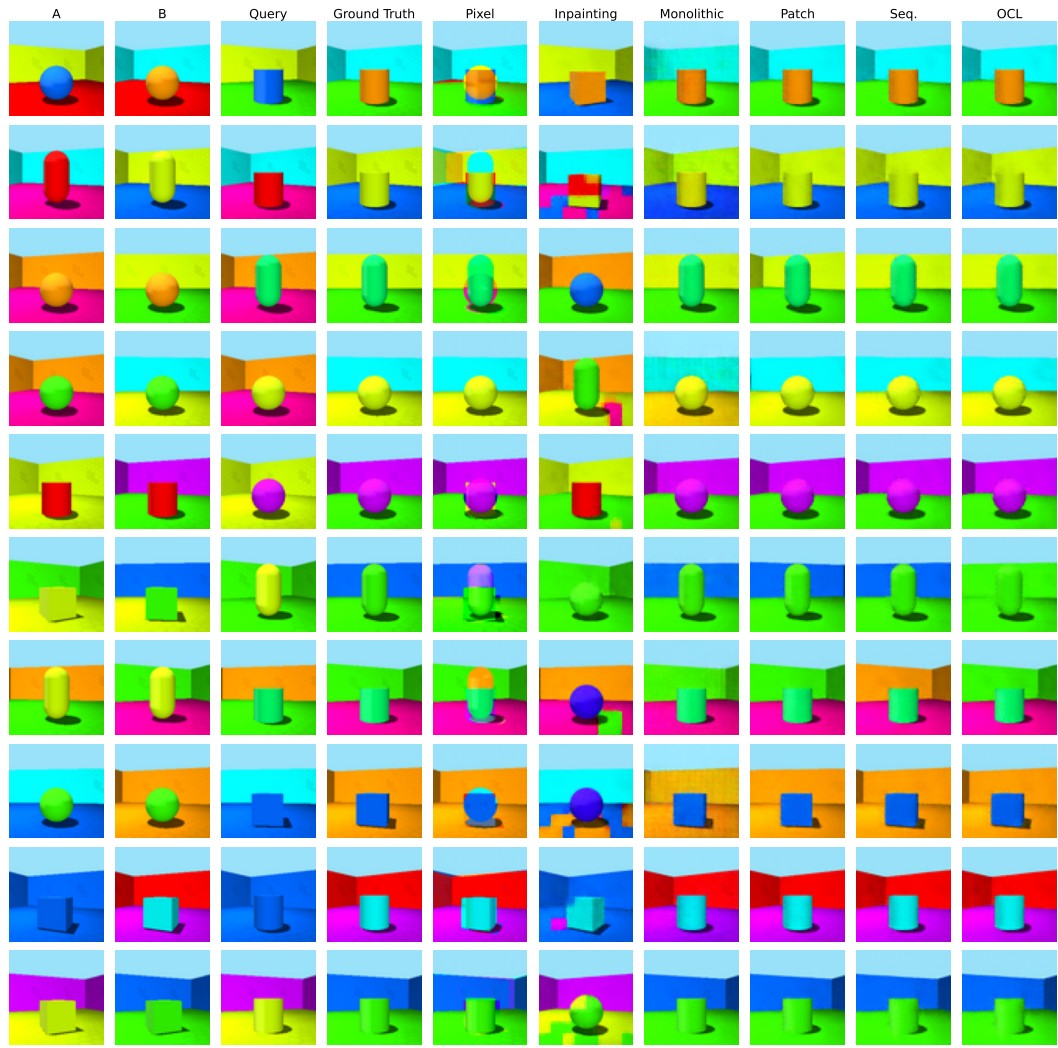

Figure 27: Three-composite task results on the 3D Shapes benchmark.

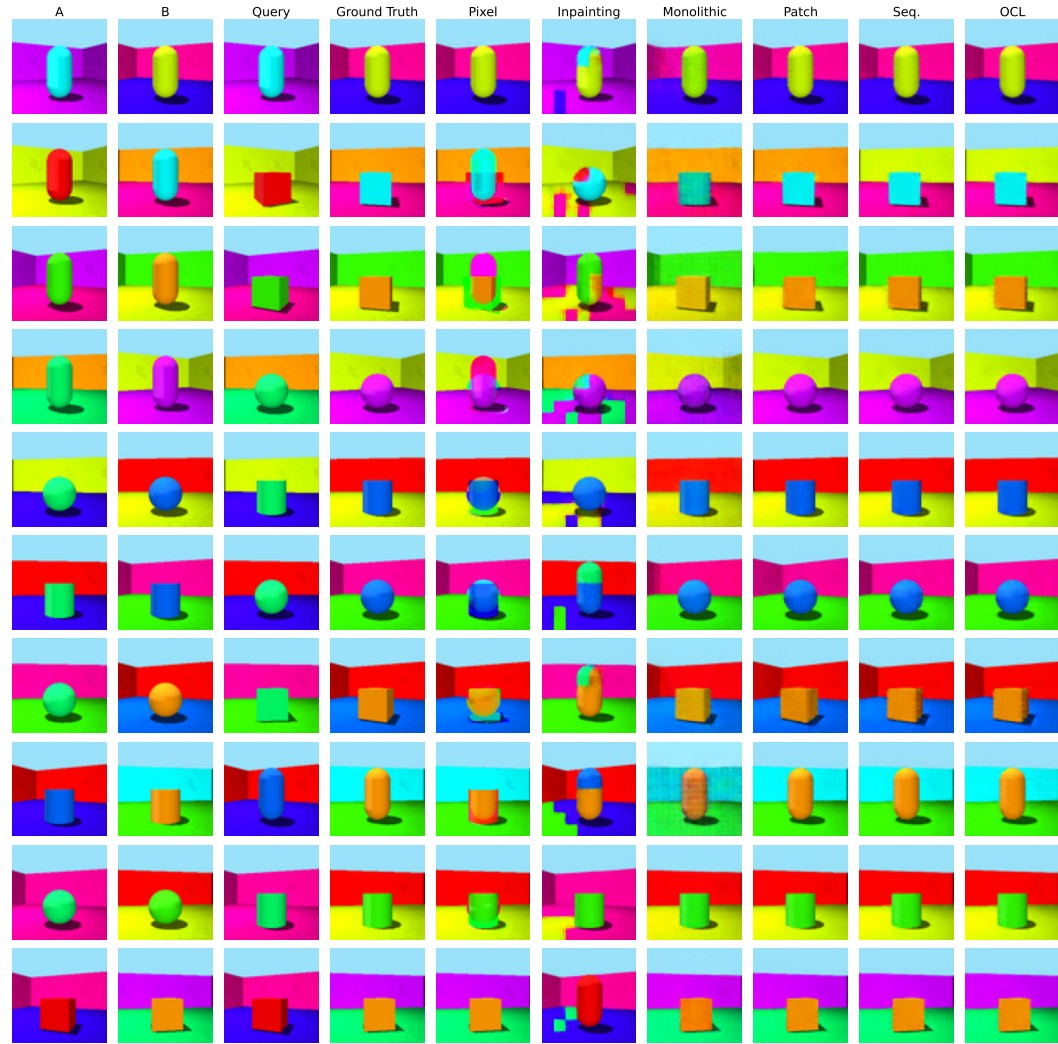

Figure 28: Four-composite task results on the 3D Shapes benchmark.

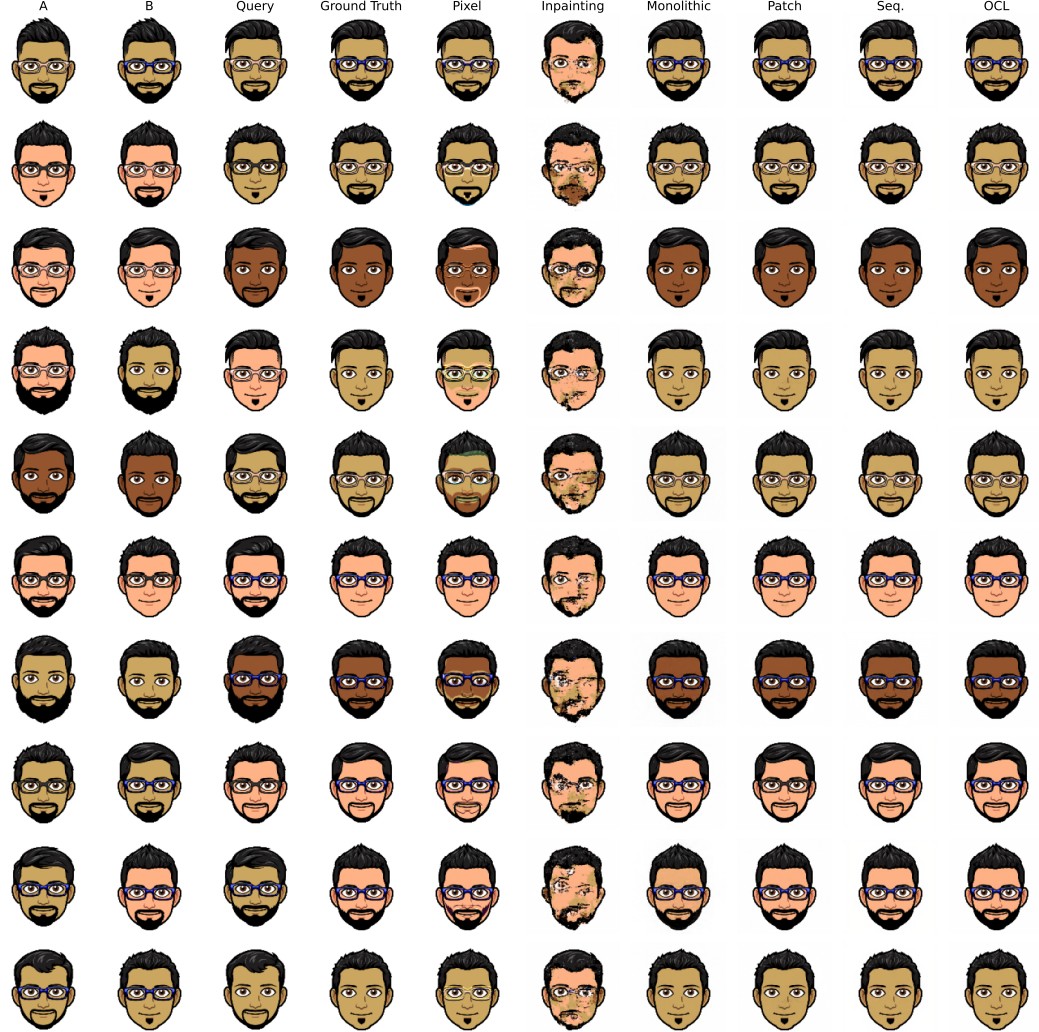

Figure 29: Two-composite task results on the BitMoji Faces benchmark.

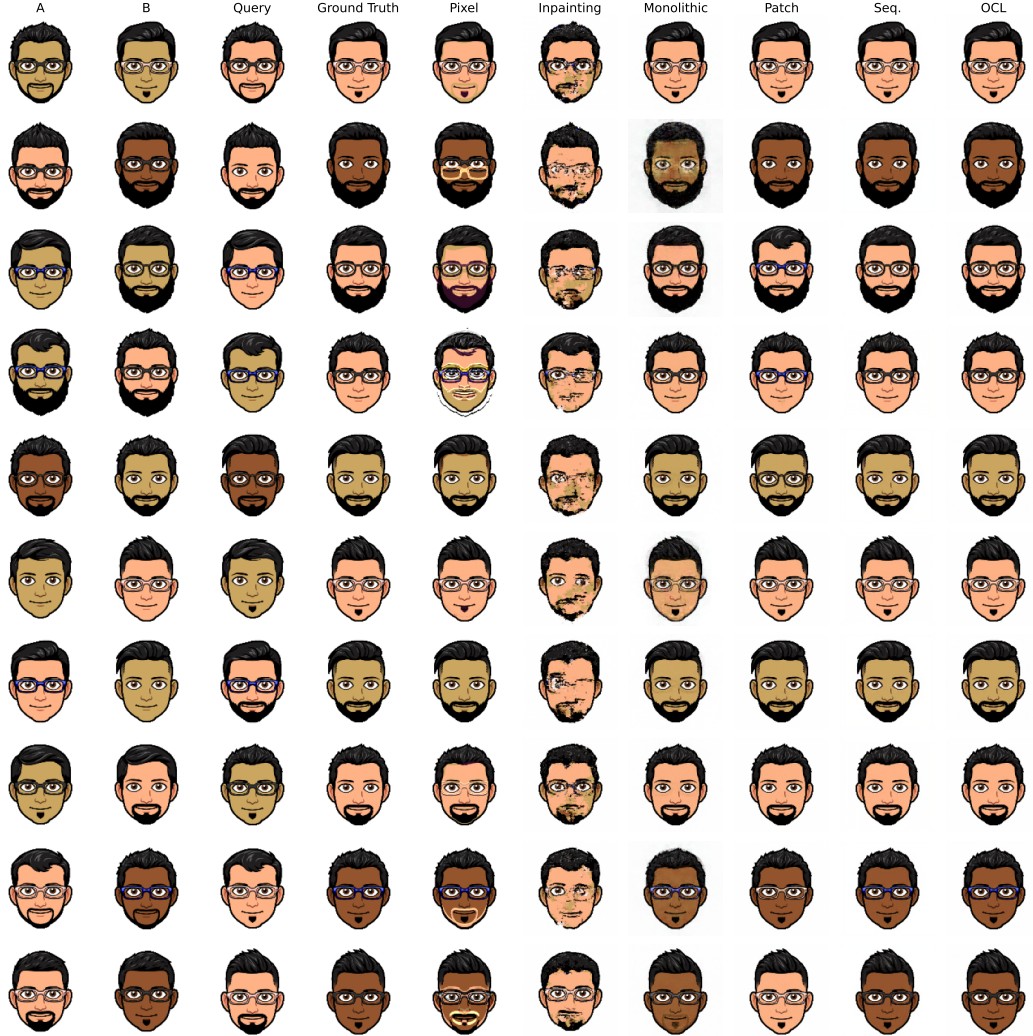

Figure 30: Three-composite task results on the BitMoji Faces benchmark.

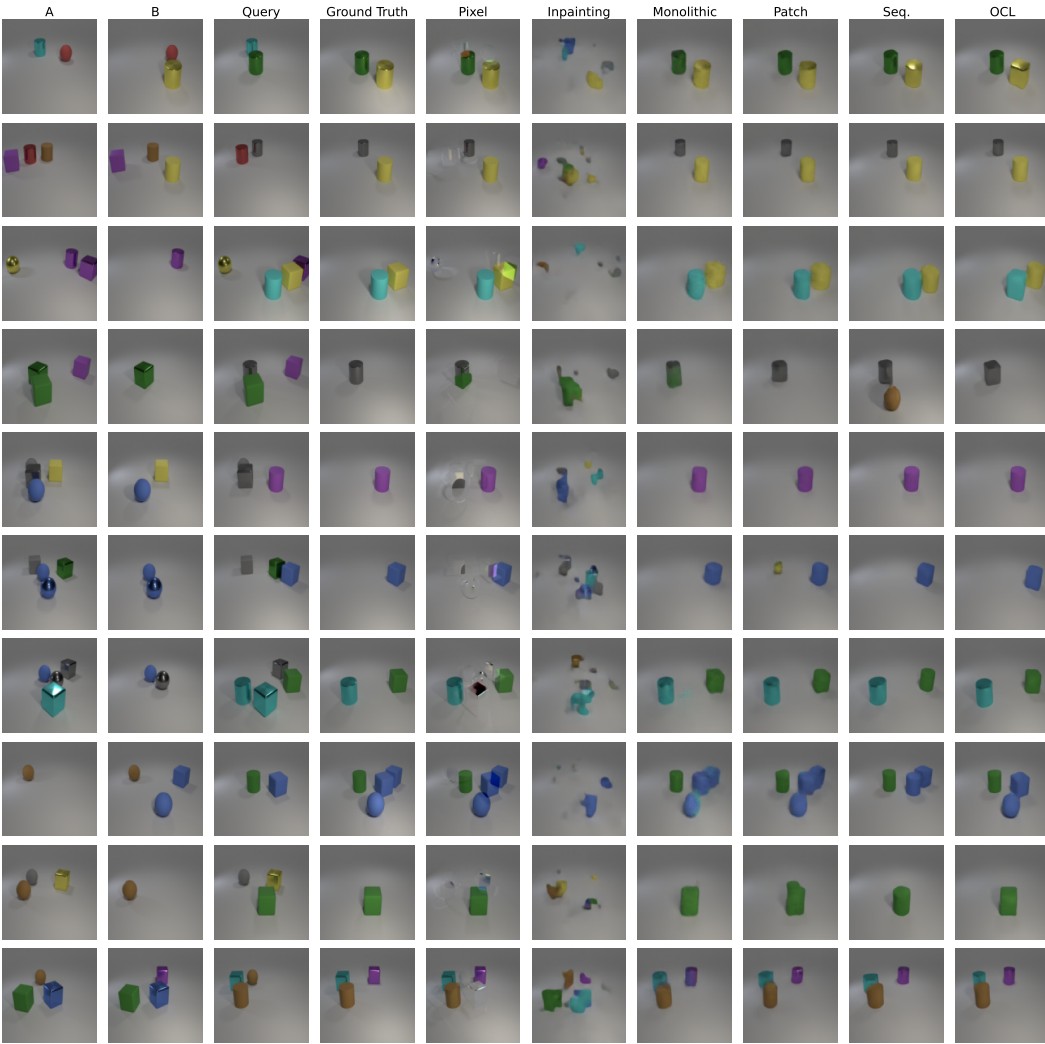

Figure 31: Two-composite task results on the CLEVr Objects benchmark.

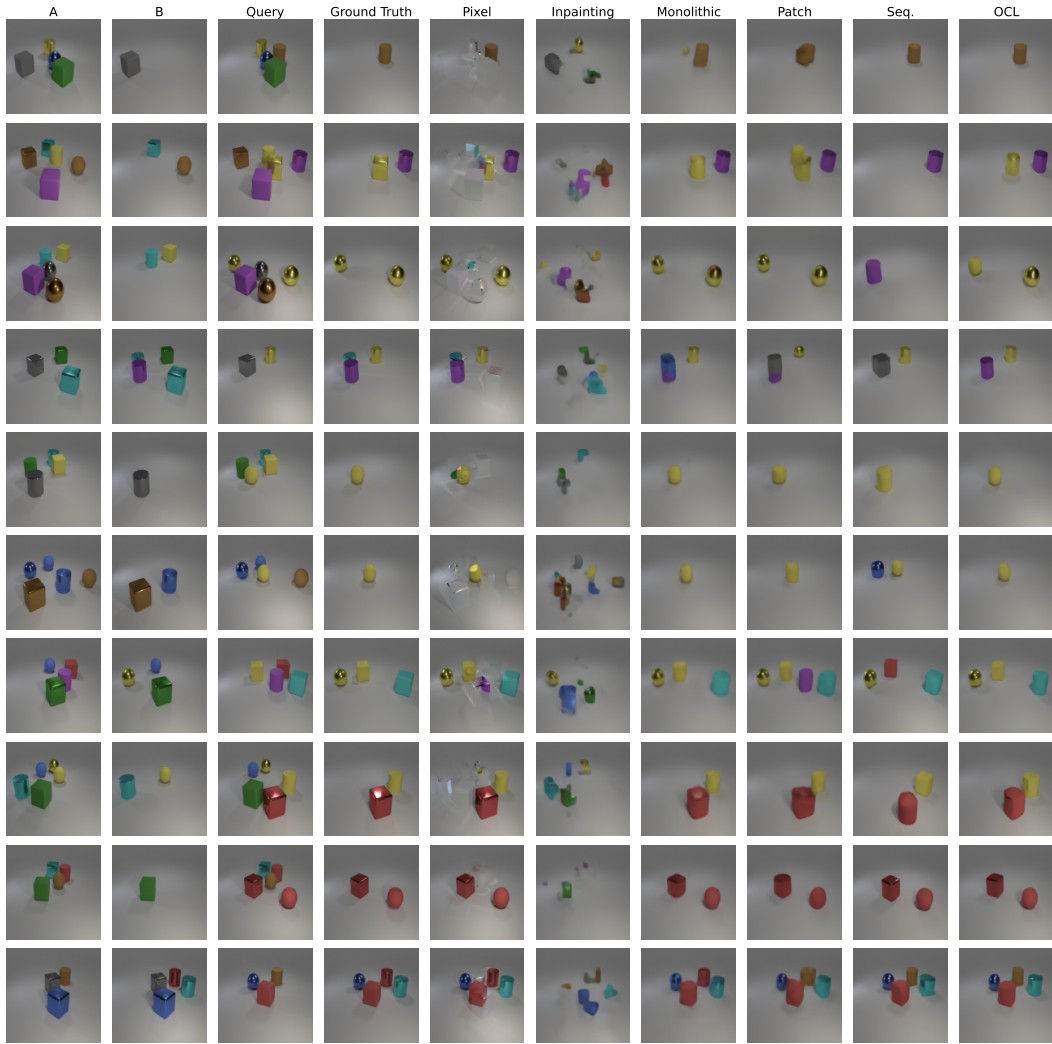

Figure 32: Three-composite task results on the CLEVr Objects benchmark.