# OpenReview forum: "Im-Promptu: In-Context Composition from Image Prompts"
_NeurIPS.cc/2023/Conference — NeurIPS 2023 poster_

### Official Review · Reviewer_JrRD · 2023-06-25

**Soundness:** 4 excellent
**Presentation:** 4 excellent
**Contribution:** 3 good
**Rating:** 8
**Confidence:** 3

**Summary:**

This paper presents a comprehensive study of in-context learning of compositionality in images. It introduces a simple benchmark of compositional language-driven visual transformations and explores under what conditions a model can perform in-context analogies from image pairs.

**Strengths:**

Very interesting study, with relevant conclusions that can be easily turned into hypotheses for examining how one can achieve in-context analogical reasoning in a more real-world, less synthetic context. The scope of the study is well delineated, and the hypotheses that are tested are very clear. This is a very important topic beyond the narrow setting of 'can we do visual analogies', because of the widespread believe that analogy making is a great proxy for broader understanding of the visual world.
This study further reinforces recent observations from a number of researchers that object or attribute-centric representations are very effective at enabling higher level visual reasoning and deserve more attention (pun intended), and confirms that cross-attention is a strong contender for enabling compositional learning beyond just a single modality.


**Weaknesses:**

One could argue that results obtained on synthetic data always come with the question of whether the findings will transfer to more real-world settings. In this case, the diversity of synthetic data sources partially addresses the concern.

Nits:
- Fig 1: girafe has two 'f' in English.
- Suggest shortening refs [17] and [19] if permitted byt the NeurIPS style guide. A whole printed page for a single reference is a bit over the top.

**Questions:**

None.

**Limitations:**

Limitations of the work are adequately spelled out.

---

> ### Author Rebuttal · Authors · 2023-08-09
>
> We thank the reviewer for the comments and positive feedback.

---

> > ### Comment · Reviewer_JrRD · 2023-08-10
> >
> > ack on rebuttal

---

### Official Review · Reviewer_FiDL · 2023-07-03

**Soundness:** 4 excellent
**Presentation:** 4 excellent
**Contribution:** 4 excellent
**Rating:** 8
**Confidence:** 4

**Summary:**

This paper focuses on the problem of using composable elements of visual modality to perform in-context referring for analogical reasoning like LLMs do. Firstly, the authors constructed three benchmarks to evaluate the generalization capacities of visual in-context learner. Then, the authors unified the formula of in-context analogy learning, and designed a framework called Im-Promptu. Different levels of agents, ranging from simple pixel-space rules to object-centric learners are trained for exploring the granularity of composition in visual modality. Lastly, the paper pointed out the characteristics of different degrees of compositionality via experiments.

**Strengths:**

1.	The paper proposes an interesting topic of considering the in-context composition learning from visual prompts. Similar to the composition skills of LLMs, visual generative models’ compositionality is largely under explored. The motivation of this paper is meaningful and insightful.
2.	This paper is of great novelty. First, it formulates the notion of analogy-based in-context learner and proposed a novel training framework called Im-Promptu based on that. What’s more, the paper provides a detailed and insightful analysis of different granularities’ influence on the compositionality and whether analogical reasoning enables in-context generalization.
3.	The paper contributes three benchmarks for evaluating the generalization properties of visual in-context learners.


**Weaknesses:**

1.	There is no obvious weakness I have found in this paper.

**Questions:**

1.	Just one question: The proposed three benchmarks are very different from actual scenes, are there any plans to introduce more real-scene images to the benchmark?

**Limitations:**

The authors discussed limitations and there is no potential negative societal impact of their work from my perspective.

---

> ### Author Rebuttal · Authors · 2023-08-09
>
> We thank the reviewer for the comments and positive feedback. We address the various concerns below.
>
> > Just one question: The proposed three benchmarks are very different from actual scenes, are there any plans to introduce more real-scene images to the benchmark?
>
> We are experimenting with photorealistic graphics engines that can render textures, near-natural objects, and finer lighting to close the sim-to-real gap.

---

> > ### Comment · Reviewer_FiDL · 2023-08-21
> >
> > Thanks authors for their rebuttal. I will keep my score. Excellent work!

---

### Official Review · Reviewer_1AXW · 2023-07-04

**Soundness:** 3 good
**Presentation:** 2 fair
**Contribution:** 2 fair
**Rating:** 4
**Confidence:** 4

**Summary:**

This work proposes a framework for in-context image generation/composition from image prompts. The in-context learner resembles analogy completion and is simply optimized with a reconstruction objective. Several variants in terms of visual representations (pixels/latent vectors), the compositionality of the vectors (monolithic/object-centric), and the scheme of integrating prompts (cross-attention/simple concatenation) are explored to indicate the key ingredients of a successful visual in-context learning model. In addition, a benchmark consisting of three object/attribute-combinatorial datasets is created to evaluate visual in-context learning tasks.

**Strengths:**

+ The main idea of introducing in-context inference ability (which is text-free) into the object-centric learning framework is interesting.

+ The proposed Object-Centric Learner (OCL) is technically sound and the experimental results demonstrate its validity for in-context composition from image prompts.

+ Extensive evaluations regarding different levels of compositionality and different means of context aggregation are investigated.


**Weaknesses:**

- The main concern comes from the confusing contribution clarification of this paper. I do not see which specific/major contributions the authors want to emphasize. Text-free in-context generation compared to text-to-image generation models such as DALL-E? Endowing SLATE with the in-context composition capability by inserting a context encoder transformer?
(a) If the answer is the first one, I would say it is over-claimed. Although Figure 6 to some extent showcases that DALL-E requires tedious prompt engineering and some external techniques like inpainting to achieve scene composition, the proposed text-free in-context generation method is not more appealing since 1) it can only be applied in the toy scenario (while DALL-E supports open-vocabulary generations with high fidelity), and 2) one needs to have each component in advance (e.g., an image of a rubber purple cube in the desired position), and this is not feasible in practical scenarios.
(b) If the answer is the latter, I would say the claimed contribution is not well supported. The proposed OCL demonstrates its ability of in-context composition from image prompts, but SLATE also supports compositional generation using the concept library built from the training datasets. I think some comparisons and discussions with SLATE would help contextualize this paper.

- Overall, the paper is not easy to follow because: 1) there are many long sentences with less specific descriptions, and 2) many technical details of the in-context learner are missing and Section 4 is redundant (e.g., Equations 2 and 3 could be moved to the appendix). In addition, the authors should consider adding some preliminaries of slot attention and SLATE, making the paper more self-contained.

- The proposed method is only evaluated on toy synthetic datasets. Although previous related works (e.g., SLATE) generally struggle to generalize to real-world data with diverse scenes, I would like to see some experiments on natural images.



**Questions:**

In terms of the experiments, I have two questions:
(a) How to implement the multi-shot in-context generation?
(b) In Figure 6, the objects in the generated image seem to be placed in the same position as they are in the image prompt. How is this position-preserving functionality realized by OCL? I would like to see some insights into this from the authors.

**Limitations:**

Yes

---

> ### Author Rebuttal · Authors · 2023-08-09
>
> We thank the reviewer for the comments and constructive feedback. We respond to the questions and concerns below.
>
> > The main concern comes from the confusing contribution clarification of this paper. I do not see which specific/major contributions the authors want to emphasize. Text-free in-context generation compared to text-to-image generation models such as DALL-E? Endowing SLATE with the in-context composition capability by inserting a context encoder transformer?
>
> The goal of the work is the latter, i.e., enabling in-context composition for visual agents, with the former being a demonstration of how compositional visual understanding could be useful for downstream tasks.
>
> >  If the answer is the latter, I would say the claimed contribution is not well supported. The proposed OCL demonstrates its ability of in-context composition from image prompts, but SLATE also supports compositional generation using the concept library built from the training datasets. I think some comparisons and discussions with SLATE would help contextualize this paper.
>
> LLMs have the uncanny ability to implicitly compose outputs using human-designed abstractions (words, BPE, sentences). On the other hand, methods like SLATE learn their own abstractions; however, composition follows via explicitly constructed concepts by the human user.  Such rules can be tedious to specify as they scale exponentially with the number of stored concepts. Our work then combines the best of both approaches by simultaneously learning the compositional abstractions and the composition process. Moreover, the implication of this goes beyond simply inserting a context encoder over SLATE. While SLATE is a state-of-the-art slot learning method, it does not a priori follow that slot abstractions can enable state-of-the-art in-context compositional generation. Our work rigorously compares slots against the other possible composition choices to test this hypothesis.
>
> > If the answer is the first one, I would say it is over-claimed. Although Figure 6 to some extent showcases that DALL-E requires tedious prompt engineering and some external techniques like inpainting to achieve scene composition, the proposed text-free in-context generation method is not more appealing since 1) it can only be applied in the toy scenario (while DALL-E supports open-vocabulary generations with high fidelity)
>
> We agree with the reviewer and do not claim to supersede DALL-E at generating open-vocabulary generations. The example shows how in-context visual understanding using the model's own concepts could facilitate relational understanding of the scene in comparison to grounding concepts in text. Future image-generation systems could then use text for high-fidelity generations of individual concepts in conjunction with architectures like OCL to facilitate the understanding of the interplay of these concepts.
>
> > The proposed method is only evaluated on toy synthetic datasets. Although previous related works (e.g., SLATE) generally struggle to generalize to real-world data with diverse scenes, I would like to see some experiments on natural images.
>
> Please refer to the section “Concern surrounding the absence of natural images” in the global response.
>
> > How to implement the multi-shot in-context generation?
>
> The context encoder collectively attends to the set of all context examples in the multi-shot case. Simple example encodings (implemented as positional embeddings) help the encoder differentiate between examples.  We will add this point to the text in addition to releasing our code, where the implementation will be made much more clear.
>
> > In Figure 6, the objects in the generated image seem to be placed in the same position as they are in the image prompt. How is this position-preserving functionality realized by OCL? I would like to see some insights into this from the authors.
>
> Note that the OCL is trained on object addition and deletion analogies (see Figure 9 in the supplementary). The image prompt then corresponds to an analogy of simply adding the object at a particular position in the scene. We will add a section in the appendix specifying the algorithm used to generate the prompts and the corresponding compositions.
>
> > Many technical details of the in-context learner are missing and Section 4 is redundant (e.g., Equations 2 and 3 could be moved to the appendix). In addition, the authors should consider adding some preliminaries of slot attention and SLATE, making the paper more self-contained.
>
> We have added detailed background for slot attention and SLATE in the Appendix (see Section D in Appendix). All technical details have been specified in Table 6 of the Appendix. If the reviewer could be more specific about the details they are looking for, we would be happy to provide the necessary information.

---

> > ### Comment · Reviewer_1AXW · 2023-08-19
> > **more comments**
> >
> > Thanks authors for their responses. The rebuttal partially addresses my concerns, but I lean towards rejection given that (1) the contribution is not well justified; (2) empowering SLATE with the ability of in-context generation via explicit image prompts is interesting but the technical contribution is limited (as pointed by Reviewer 4dpY); (3) the evaluation on toy datasets is not adequate in my view; (4) the writing and presentation need substantial improvements. Thanks.

---

> > > ### Author Response · Authors · 2023-08-20
> > > **Reply to response**
> > >
> > > Thank you for your constructive feedback. We believe that our work provides a crucial bottom-up understanding of compositional generalization from few-shot examples that can further help understand in-context learning in modalities beyond language. We encourage you to parse through the response to reviewer 4dpy for a detailed enumeration and implications of key contributions. In addition, our experiments also show that "visual analogy" methods pointed out by reviewer 4dpy (Ref [1] and [3]) exhibit limited generalization to higher-order composition prompts.  Our datasets, although synthetic, consist of benchmarks of varying complexity and semantic diversity. While datasets based on real images are able to measure task-specific metrics, we provide an avenue for systematically testing and measuring the broader compositional generalization limits of future methods.

---

### Official Review · Reviewer_Lzbc · 2023-07-24

**Soundness:** 3 good
**Presentation:** 3 good
**Contribution:** 3 good
**Rating:** 7
**Confidence:** 4

**Summary:**

This work investigates whether analogical reasoning can enable in-context composition over composable elements of visual stimuli. First, they introduce a suite of three benchmarks to test the generalization properties of a visual in-context learner. They formalize the notion of an analogy-based in-context learner and use it to design a meta-learning framework called Im-Promptu. To this end, they use Im-Promptu to train multiple agents with different levels of compositionality, including vector representations, patch representations, and object slots. Lastly, they demonstrate a use case of Im-Promptu as an intuitive programming interface for image generation.

**Strengths:**

In-context learning is a novel learning paradigm recently. While in-context learning is popular with large language models (LLM), it has not been discovered in computer vision fields. This paper provides three benchmarks for in-context learning for computer vision and introduces some algorithms for proposed benchmarks. Since in-context learning is open for computer vision, the attempt is valuable and makes further research possible.

**Weaknesses:**

In-context learning is a general learning paradigm for diverse tasks. In the computer vision field, there are a lot of tasks (e.g., classification, object detection, segmentation). However, this paper only considers three tasks: 3D Shapes Image Prompts, BitMoji Faces Image Prompts, CLEVr Objects Image Prompts. Thus, this benchmark may be not sufficiently comprehensive from this perspective. On the other hand, the performance versus the number of in-context examples is one important aspect of in-context learning. This paper doesn't provide any insight from this view.

**Questions:**

I have the following questions:
1. Since the number of in-context demonstrations is one important parameter in in-context learning fields, is there any ablation study for the number of demonstrations?
2. The input and output of these three tasks are all images. One question is can the text be the output of the task (e.g., image classification)?
3. In the proposed method, there are some components involved. What's the difference if we use models from scratch instead of pre-trained?


**Limitations:**

Compared with in-context learning in language models, this paper only considers three proposed computer vision tasks. Thus the setting of in-context learning may be limited.

---

> ### Author Rebuttal · Authors · 2023-08-09
>
> We thank the reviewer for the comments and positive feedback. We address the various concerns below.
>
> > Since the number of in-context demonstrations is one important parameter in in-context learning fields, is there any ablation study for the number of demonstrations?
>
> We agree with the reviewer that the number of context examples is a critical factor to in-context learning ability. We have, in fact, run ablations on the number of examples by exploring k-shot variants of the models. The plots in Figure 4 and Figure 5  show the model performance against the number of examples under “k-shot” labels. In addition, observations on the effect of context size have been described in R1.3.  We find the performance of the sequential prompter and patch-based agent to significantly improve with additional examples.
>
> > The input and output of these three tasks are all images. One question is can the text be the output of the task (e.g., image classification)?
>
> We think this is possible by jointly modeling slots and language tokens and doing the context aggregation over the joint space.
>
> > In the proposed method, there are some components involved. What's the difference if we use models from scratch instead of pre-trained?
>
> As such, we found that symmetry breaking among slots converged extremely slowly when trained from scratch and was sensitive to hyperparameters. We found it helpful to pre-train the encoder and bias the slots to capture object abstractions before learning to compose.

---

> > ### Comment · Reviewer_Lzbc · 2023-08-19
> > **Post-rebuttal**
> >
> > After reading other reviewers' reviews and authors' rebuttals, I decide to keep my positive score. Even though the setting in the paper is not comprehensive, I still think in-context learning is an interesting topic in computer vision and this paper can provide benchmarks and some insights for the community.

---

### Official Review · Reviewer_4dpY · 2023-07-26

**Soundness:** 2 fair
**Presentation:** 2 fair
**Contribution:** 1 poor
**Rating:** 4
**Confidence:** 4

**Summary:**

The paper investigates whether analogical reasoning can enable in-context compositional generalization over visual entities. The authors construct three benchmarks to test the compositional generalization on visual analogy-making, including 3D shapes, BitMoji Faces, and CLEVR. The authors also present a visual analogy-making algorithm called Im-Promptu. The authors test the proposed method on the constructed benchmarks with various visual representations, including vector representations, patch representations, and object slots. The experiments demonstrate the tradeoffs between extrapolation abilities and the compositionality degree of the visual representations.

**Strengths:**

1. The paper constructs three benchmarks to test the compositional generalization on visual analogy-making, including 3D shapes, BitMoji Faces, and CLEVR.

2. The paper conducts experiments using various visual representations, including vector representations, patch representations, and object slots, and provides insights into the impact of the compositionality degree of the visual representations on compositional generalization. The results demonstrate the effectiveness of the object-centric representation for the compositional visual analogy-making.

**Weaknesses:**

1. The technical novelty of the proposed framework (Im-Promptu) is limited. The formulation in Section 4 is similar to visual analogy-making [1,2]. It is inappropriate to rename it as in-context learning or Im-Promptu learning, if there is no fundamental difference. Section 5 lists several model variants for visual analogy-making by composing existing modules. It is unclear what is the paper's contribution to the methodology.

2. The constructed benchmarks are all synthetic domains with little variance. This is a weakness considering that previous works [2,3] have already studied similar tasks using natural images.

3. The experimental results are inadequate. The authors only evaluated the proposed framework on the self-constructed benchmarks, while previous works [1,2,3] have introduced several benchmarks for visual analogy-making. It is unclear whether the proposed framework generalizes to other realistic benchmarks.

4. The paper lacks reference to important related works [2,3].

[1] Reed, Scott E., et al. "Deep visual analogy-making." Advances in neural information processing systems 28 (2015).

[2] Sadeghi, Fereshteh, C. Lawrence Zitnick, and Ali Farhadi. "Visalogy: Answering visual analogy questions." Advances in Neural Information Processing Systems 28 (2015).

[3] Bar, Amir, et al. "Visual prompting via image inpainting." Advances in Neural Information Processing Systems 35 (2022): 25005-25017.


**Questions:**

1. What is the fundamental difference between the proposed Im-Promptu learning in Section 4 and visual analogy-making?

2. What is the main technical contribution of this paper?

3. The method of this paper works better on spatially consistent datasets, so how to reflect the statement "produces a more generalized composition beyond spatial relations" in the paper?

4. Why are there some results with lower MSE and higher FID?

**Limitations:**

The authors do not discuss the limitations of the proposed method.

---

> ### Author Rebuttal · Authors · 2023-08-09
>
> We thank the reviewer for the comments and constructive feedback. We respond to the questions and concerns below.
>
> > What is the fundamental difference between the proposed Im-Promptu learning in Section 4 and visual analogy-making?
>
> We agree that our work bears similarity to visual analogy-making in the sense of being able to complete an analogy, having observed a transformation over context examples. However, our work is interested in asking the broader question of what enables the in-context compositional generation over composable attributes in the visual domain. Such a compositional learning ability has been well investigated for the language modality. This question can then be disentangled into the interplay of three sub-questions - (1) What composable abstractions can help learn implicit composition? (2) Which objective optimizes learning such rules? (3) How to model the composition from entities? In this context, visual analogy as an auto-encoding reconstruction only answers the second question by formulating a reliable optimization objective via analogy solving for learning transformations. Im-Promptu, on the other hand, supports broader compositional learning by unifying these components under a common mechanistic framework (See Equations 1 and 3).
>
> >The technical novelty of the proposed framework (Im-Promptu) is limited. The formulation in Section 4 is similar to visual analogy-making [1,2,3]. It is inappropriate to rename it as in-context learning or Im-Promptu learning if there is no fundamental difference.
>
> First, Ref. [2] focuses on solving intelligence test types of analogies where the solver has to classify the correct answer from a bunch of different options, and as such, doesn't support generation. Ref. [1] uses visual analogy as a proxy to learn input-invariant transformation vectors that can traverse continuous manifolds (e.g., geometric transformations, animation frames) via simple rules. The authors demonstrate limited generalization to new inputs but with latent transformations within the training distribution. In fact, our monolithic agent is directly instantiated from their MLP formulation (See Equation 5.3 in Ref. [1]). Our experiments reinforce the findings of [1] and demonstrate the ability of such representations to generalize to single-order transformations (Sec. 6.2) at test time. However, these don't extend to a higher-order compositional generation where multiple composite attributes are modified and therefore don’t support higher-order visual understanding (Sec. 6.3).
>
> We would like to thank the reviewers for pointing us toward the in-painting technique used in [3]. We have now implemented the technique with the slight modification of using an MAE -VQVAE ensemble (instead of MAE-VQGAN). The pretraining involves masked auto-encoder (MAE) reconstruction on 2x2 analogy grids described in [3]. See the global response section for exact quantitative and qualitative results. Even though past works have found in-painting as a simple and effective self-supervision task for learning general-purpose representations, we find that it cannot generate compositions beyond the training distribution (see attached pdf for generated outputs).
>
> > Section 5 lists several model variants for visual analogy-making by composing existing modules. It is unclear what is the paper's contribution to the methodology.
>
> We believe that our work takes a crucial step towards solving visual analogies that can understand and compose higher-order relations than the ones they were trained on, a notion largely unexplored in visual analogy models. To this end, we make several key technical contributions:
>
> 1. Benchmarks for the systematic understanding of in-context learning in the visual domain. This is important to understand the underpinnings of in-context learning, the emergence of which is less than completely understood in the language domain.
> 2. Extensive exploration of model abstractions across the compositionality continuum. While we use existing modules to learn compositional abstractions, it is not a priori straightforward that strong compositional abstractions should also support in-context compositional generation. Ours is the first work that utilizes these modules with a range of context aggregation techniques to understand the implications of various design choices on generalization.
> 3. From a more practical standpoint, we provide empirical training recipes for OCL that could be utilized to scale it while doing in-context learning over real data.
>
> > The constructed benchmarks are all synthetic domains with little variance. This is a weakness considering that previous works [2,3] have already studied similar tasks using natural images. It is unclear whether the proposed framework generalizes to other realistic benchmarks
>
> Please see the global response section under the heading “Concern surrounding the absence of natural images”. While other works [3] that use realistic benchmarks show decent performance across task-specific metrics, the degree of generalization that can be obtained via in-painting is, as such, unmeasurable. In fact, our experiments show that the techniques of both Refs. [1] and [3] are not amenable to compositional generalization.
>
> > The method of this paper works better on spatially consistent datasets, so how to reflect the statement "produces a more generalized composition beyond spatial relations" in the paper?
>
> Here we refer to the aforementioned papers' limited ability to generalize to spatial “positional” relations by separating the position of an object from its content. Our method, in contrast, can be used to simultaneously modify the geometry of the scene with other compositional attributes. We apologize for the confusion.
>
> >  Why are there some results with lower MSE and higher FID?
>
> This is possible if the model generation gets the larger structure of the output right but poorly captures finer aspects like shadows and edge transitions.

---

> > ### Comment · Reviewer_4dpY · 2023-08-21
> >
> > Thank the authors for the response and additional results. While the rebuttal address part of my concerns, I am still not convinced that the so-called Im-Promptu learning is fundamentally different from visual analogy-making. Besides, the proposed method is of limited novelty, and the proposed benchmarks are synthetic and limited. Therefore, I keep my rating towards rejection.

---

### Author Rebuttal · Authors · 2023-08-09

We would like to thank all reviewers for raising a number of great questions, adding detailed comments, and drawing our attention to related works. We will incorporate the feedback in the next iteration of the manuscript. We address some common concerns below and introduce additional results.

### Concern surrounding the absence of natural images

Formulating a method that can solve in-context downstream computer vision tasks is a worthwhile goal, but the scope of this work lies in understanding the computational underpinnings of compositional generalization. Uncovering the process by which humans form and modify flexible representations from few-shot examples is key to abstract reasoning and visual understanding [1][2]. While training large models on natural images is an appealing avenue for studying the emergence of large-scale in-context learning, this by itself tells us very little about the acquisition and the generalization limits of composition skills. Therefore, in this work, we construct synthetic benchmarks with composable attributes to systematically find effective inductive biases for compositional generalization.  We believe that the key to scaling up our models lies in designing natural image benchmarks where compositional attributes of the visual stimuli are modified. Generating compositional stimuli for real images is a significant engineering effort, and we leave this to future work.

### Additional Inpainting Results

We thank Reviewer 4dpy for pointing us to recent work [3] on visual prompting through in-painting. The authors represent an analogy via an image grid and use inpainting to fill in the missing solution. We implemented this methodology to investigate its ability to acquire composition skills. The results are as follows:

**Primitive Task Extrapolation**:

|           | OCL-1 shot MSE | In-Painting MSE | OCL-1 shot FID | In-Painting FID |
| --------- | ----------------- | --------------- | ----------------- | --------------- |
| Shapes 3D | 4.72 ∓ 0.46       | 142.77 ∓ 1.31   | 34.77 ∓ 0.06      | 31.20 ∓ 1.86    |
| BitMoji   | 6.91 ∓ 0.54       | 260.94 ∓ 8.27   | 8.96 ∓ 0.05       | 104.84 ∓ 12.23  |
| CLEVr     | 37.99 ∓ 1.91      | 200.64 ∓ 3.75   | 38.99 ∓ 0.57      | 186.34 ∓ 5.33   |

**Composite Task Extrapolation**:

|                             | OCL-1 shot MSE   | In-Painting MSE | OCL-1 shot FID   | In-Painting FID |
| --------------------------- | ------------ | ------------------ | ------------ | ------------------ |
| Shapes 3D - Two Composite   | 6.22 ∓ 2.76  | 194.6 ∓ 1.56       | 34.18 ∓ 0.02 | 31.82 ∓ 2.45       |
| Shapes 3D - Three Composite | 7.96 ∓ 2.93  | 268.6 ∓ 0.92       | 35.12 ∓ 0.05 | 41.84 ∓ 1.51       |
| Shapes 3D - Four Composite  | 12.62 ∓ 4.41 | 292.84 ∓ 1.12      | 36.82 ∓ 0.06 | 51.34 ∓ 1.97       |
| BitMoji - Two Composite     | 5.72 ∓ 0.18  | 284.9 ∓ 7.35       | 8.78         | 103.74 ∓ 10.54     |
| BitMoji - Three Composite   | 5.96 ∓ 0.23  | 311.1 ∓ 6.48       | 8.74         | 108.92 ∓ 9.71      |
| CLEVr - Two Composite       | 37.12 ∓ 1.16 | 235.66 ∓ 2.77      | 68.45 ∓ 2.32 | 191.72 ∓ 3.85      |
| CLEVr - Three Composite     | 62.26 ∓ 1.41 | 285.76 ∓ 2.43      | 58.5 ∓ 1.02  | 183.94 ∓ 4.27      |


Examples of the generated outputs have been shown in the attached pdf. We observe that in-painting can extend composition rules to in-distribution primitives but produces incoherent outputs when tested on both primitive rule extrapolation (Sec. 6.2) and composite rule extrapolation (Sec. 6.3). It performs the worst out of all the agents. The technical details, results, and code for the In-Painting will be added to the updated manuscript.

[1] Brenden Lake et al., “Building machines that learn and think like people,” Behavioral and Brain Sciences, 2016

[2] Brenden Lake et al., “One shot learning of simple visual concepts,” Cognitive Science, 2011

[3] Amir Bar et al., "Visual prompting via image inpainting," Advances in Neural Information Processing Systems 35 (2022): 25005-25017

---

### Decision · Program_Chairs · 2023-09-21

**Decision:**

Accept (poster)

**Comment:**

This work receives mixed reviews. Three expert reviewers think the in-context (composition) learning in vision modality is important yet highly under-explored, and the contributions (synthetic benchmark and datasets, problem formulation and the meta-learning framework, evaluation and analysis of compositionality of various degrees) in this paper are significant, while the other two expert reviewers concern about its novelty against visual analogy-making and its use in realistic setting. Overall, AC agrees this work is interesting towards visual in-context learning, emphasizes more on the (high-order) compositional generalization compared to previous visual analogy-making, and can be much strenghened by  evaluating on real-world problems (e.g. [1]).

[1] Amir Bar et al., "Visual prompting via image inpainting," Advances in Neural Information Processing Systems 35 (2022): 25005-25017